# Finding Order in Chaos: A Novel Data Augmentation Method for Time Series in Contrastive Learning

**Berken Utku Demirel**
Department of Computer Science
ETH Zürich, Switzerland
berken.demirel@inf.ethz.ch

**Christian Holz**
Department of Computer Science
ETH Zürich, Switzerland
christian.holz@inf.ethz.ch

## Abstract

The success of contrastive learning is well known to be dependent on data augmentation. Although the degree of data augmentations has been well controlled by utilizing pre-defined techniques in some domains like vision, time-series data augmentation is less explored and remains a challenging problem due to the complexity of the data generation mechanism, such as the intricate mechanism involved in the cardiovascular system. Moreover, there is no widely recognized and general time-series augmentation method that can be applied across different tasks. In this paper, we propose a novel data augmentation method for quasi-periodic time-series tasks that aims to connect intra-class samples together, and thereby find order in the latent space. Our method builds upon the well-known mixup technique by incorporating a novel approach that accounts for the periodic nature of non-stationary time-series. Furthermore, by controlling the degree of chaos created by data augmentation, our method leads to improved feature representations and performance on downstream tasks. We evaluate our proposed method on three time-series tasks, including heart rate estimation, human activity recognition, and cardiovascular disease detection. Extensive experiments against state-of-the-art methods show that the proposed approach outperforms prior works on optimal data generation and known data augmentation techniques in the three tasks, reflecting the effectiveness of the presented method. Source code: https://github.com/eth-siplab/Finding_Order_in_Chaos.

## 1 Introduction

Self-supervised learning methods have gained significant attention as they enable the discovery of meaningful representations from raw data without explicit annotations. These self-supervised methods learn representations without labels by designing pretext tasks that transform the unsupervised representation learning problem into a supervised one such as predicting the rotation of images [1], or contexts [2, 3]. Among these methods, contrastive learning (CL), which learns to distinguish semantically similar examples over dissimilar ones, stands out as a powerful approach in self-supervised learning across various domains including computer vision [4–6], speech recognition [7–10], and natural language processing [11–14].

The success of contrastive learning relies on the creation of similar and dissimilar examples, which is typically achieved through the use of data augmentations [15, 16]. Recently, it was shown that data augmentations have a role to create a "*chaos*" between different intra-class samples such that they become more alike. For example, two different cars become very similar when they are both cropped to the wheels. [17]. However, in time-series data, creating similar samples with augmentation techniques is more challenging due to the complexity of the dynamical data generation mechanisms [18]. Moreover, research on contrastive learning for time series has demonstrated the

37th Conference on Neural Information Processing Systems (NeurIPS 2023).

absence of a unique data augmentation technique that consistently performs better than others in different tasks [19, 20]. Instead, the choice of augmentation depends on the contextual characteristics of the signal, such as perturbing the high-frequency content of a signal that carries characteristic information in low frequencies does not generate useful data samples that are helpful for contrastive learning to learn class invariant features [21].

Considering these limitations, in this work, we first propose a novel data augmentation method for time series data by performing a tailored mixup while considering the phase and amplitude information as two separate features. Then, we perform specific operations for both features to generate positive samples by controlling the mixup coefficients for each feature to prevent aggressive augmentation. Specifically, our method employs a technique that controls the mixup ratio for each randomly chosen pair based on their distance in the latent space which is acquired through the use of a variational autoencoder (VAE), whose objective is to learn disentangled representations of data without labels. To this end, subjecting to the distance constraint in the latent space, the mixup tries to connect semantically closer samples together more aggressively while preventing the excessive interpolation of dissimilar samples that are likely to belong to different classes. Therefore, the purpose of our proposed method for quasi-periodic time-series data augmentation is to find an order in "chaos" between different samples such that they become more alike by interpolating them in a novel manner to prevent the loss of information. We summarize our contributions as follows:

- We propose a novel mixup method for non-stationary quasi-periodic time-series data by considering phase and magnitude as two separate features to generate samples that enhance intra-class similarity and help contrastive learning to learn class-separated representations.

- We present a novel approach for sampling mixup coefficients for each pair based on their similarity in the latent space, which is constructed without supervision while learning disentangled representations, to prevent aggressive augmentation between samples.

- We show that the tailored mixup with coefficient sampling consistently improves the performance of contrastive learning in three time-series tasks compared to prior mixup techniques and proposed augmentation methods that generate optimal/hard positives or negatives.

## 2 Preliminaries

### 2.1 Notation

We use lowercase symbols $(x)$ to denote scalar quantities, bold lowercase symbols $(\mathbf{x})$ for vector values, and capital letters $(X)$ for random variables. Functions with a parametric family of mappings are represented as $f_\theta(.)$ where $\theta$ is the parameters. The discrete Fourier transformation of a real-valued time series sample is denoted as $\mathcal{F}(.)$, yielding a complex variable as $X_k$ where $X \in \mathbb{C}$ and $k \in [0, f_s/2]$ is the frequency with the maximum value of Nyquist rate. The amplitude and phase of the $\mathcal{F}(\mathbf{x})$ are represented as $A(\mathbf{x})$ and $P(\mathbf{x})$. The real and imaginary parts of a complex variable are shown as $\mathrm{Re}(.)$ and $\mathrm{Im}(.)$. The detailed calculations for operations are given in the Appendix A.1.

### 2.2 Setup

We follow the common CL setup as follows. Given a dataset $\mathcal{D} = \{(\mathbf{x}_i)\}_{i=1}^{K}$ where each $\mathbf{x}_i$ consists of real-valued sequences with length L and $C$ channels. The objective is to train a learner $f_\theta$ which seeks to learn an invariant representation such that when it is fine-tuned on a dataset $\mathcal{D}_l = \{(\mathbf{x}_i, \mathbf{y}_i)\}_{i=1}^{M}$ with $M \ll K$ and $\mathbf{y}_i \in \{1, \ldots, N\}$, it can separate samples from different classes.

### 2.3 Motivation

As stated by prior works, mixup-based methods have poor performance in domains where data has a non-fixed topology, such as trees, graphs, and languages [22, 23]. Here, we demonstrate how we derived our proposed method by revealing the limitations of mixup for time series theoretically while considering the temporal dependencies and non-stationarities.

**Assumption 2.1** (SNR Matters). *There exist one or multiple bandlimited frequency ranges of interest $f^*$, where the information that average raw time-series data conveys about the labels (i.e., $\mathcal{I}(\mathbf{y}; \mathbf{x})$) is directly proportional to normalized signal power in that frequency range as in Equation 1.*

$$\mathcal{I}(\mathbf{y}; \mathbf{x}) \propto \int_{f^*} S_x(f) \, / \int_{-\infty}^{\infty} S_x(f) \quad \text{where} \quad S_x(f) = \lim_{N \to \infty} \frac{1}{2N} \left| \sum_{n=-N}^{N} x_n e^{-j2\pi fn} \right|^2 \quad (1)$$

Assumption 2.1 states that the information from a time series depends on its signal-to-noise ratio (SNR). Prior works showed that specific frequency bands hold inherent information about the characteristics of time series, which helps classification [21, 24].

**Assumption 2.2.** *The true underlying generative process $f(.)$, for a given data distribution $\mathcal{D} = \{\mathbf{x}_k\}_{k=1}^K$, is quasiperiodic, i.e., $f(\mathbf{x} + \tau) = g(\mathbf{x}, f(\mathbf{x}))$, where $\tau$ can be either fixed or varied.*

Assumption 2.2 posits that the observed data samples from the distribution $\mathcal{D}$ are generated by a quasiperiodic function. This is a minimal assumption since the quasiperiodicity is the relaxed version of the periodic functions. In simpler terms, quasiperiodicity can be described as the observed signals exhibiting periodicity on a small scale, while being unpredictable on a larger scale. And, several prior works showed that the data generation mechanism of time-series data for several applications in the real world are quasiperiodic [25–30]. Therefore, Assumption 2.2 is realistic.

**Proposition 2.3** (Destructive Mixup). *If Assumptions 2.1 and 2.2 hold, there exist $\lambda \sim Beta(\alpha, \alpha)$ or $\lambda \sim U(\beta, 1.0)$ with high values of $\beta$ such that when linear mixup techniques are utilized, the lower bound of the mutual information for the augmented sample distribution decreases to zero.*

$$0 \leq \mathcal{I}(\mathbf{y}; \mathbf{x}^+) < \mathcal{I}(\mathbf{y}; \mathbf{x}^*), \quad \text{where } \mathbf{x}^* \text{is the optimal sample,}$$
$$\mathbf{x}^+ = \lambda \mathbf{x} + (1 - \lambda)\tilde{\mathbf{x}} \quad \text{and} \quad \int_{f^*} S_{x^*}(f) = \int_{-\infty}^{\infty} S_{x^*}(f) \quad (2)$$

Proofs can be found in Appendix A. This proposition indicates that although the augmented samples are primarily derived from anchor samples ($\mathbf{x}$) with high ratios, the resulting instances may not contain any task-relevant information. In other words, the augmentation process can potentially discard the whole task-specific information. This destructive behavior of mixup for quasi-periodic data can be attributed to the *interference* phenomenon in which two waves interact to form the resultant wave of the lower or higher amplitude according to the phase difference as shown in Proposition 2.3.

## 3 Method

We introduce a novel approach to overcome the limitations of mixup by treating the magnitude and phase of sinusoidal signals as two distinct features with separate behaviors. Subsequently, we apply tailored mixup operations to each feature, considering their specific characteristics and effects.

We perform the linear mixup for the magnitude of each sinusoidal. However, for the phase, we take a different approach and bring the phase components of the two coherent signals together by adding a small value to the anchor's phase in the direction of the other sample. The mixup operation performs the linear interpolation of features [31], however, interpolation of two complex variables can result in a complex variable whose phase and magnitude are completely different/far away from those two, i.e., mixup can be destructive extrapolation rather than the interpolation of features. Therefore, we mix the phase of two sinusoidal as follows. We start by calculating the shortest phase difference between the two samples, denoted as $\Delta\Theta$, as described in Equation 3[1].

$$\theta \equiv [P(\mathbf{x}) - P(\tilde{\mathbf{x}})] \pmod{2\pi}$$
$$\Delta\Theta = \begin{cases} \theta - 2\pi, & \text{if } \theta > \pi \\ \theta, & \text{otherwise} \end{cases} \quad (3)$$

The sign of the calculated phase difference provides information about the relative phase location of the other sample, in either a clockwise or counterclockwise direction in the phasor diagram. And, the absolute value of it represents the shortest angular difference between two samples in radians. Based

---

[1]We use phase in radians throughout the paper in the range $(-\pi, \pi]$

on the calculated phase difference between two samples, we perform mixup operation to generate diverse positive samples as in Equation 4 such that phase and magnitude of augmented instances are interpolated properly according to the anchor sample $\mathbf{x}$, without causing any destructive interference.

$$\mathbf{x}^+ = \mathcal{F}^{-1}(A(\mathbf{x}^+)\angle P(\mathbf{x}^+)) \quad \text{where} \ \ A(\mathbf{x}^+) = \lambda_A A(\mathbf{x}) + (1-\lambda_A)A(\tilde{\mathbf{x}}) \ \ \text{and}$$

$$P(\mathbf{x}^+) = \begin{cases} P(\mathbf{x}) - |\Delta\Theta| * (1-\lambda_P), & \text{if } \Delta\Theta > 0 \\ P(\mathbf{x}) + |\Delta\Theta| * (1-\lambda_P), & \text{otherwise} \end{cases} \tag{4}$$

The proposed method which mixes the magnitude and phase of each frequency component with tailored operations, not only prevents destructive interference between time series, resulting in an increase in the lower bound of mutual information (as shown in Theorem 3.1), but also generates diverse augmented instances with the same two samples by using two different mixing coefficients.

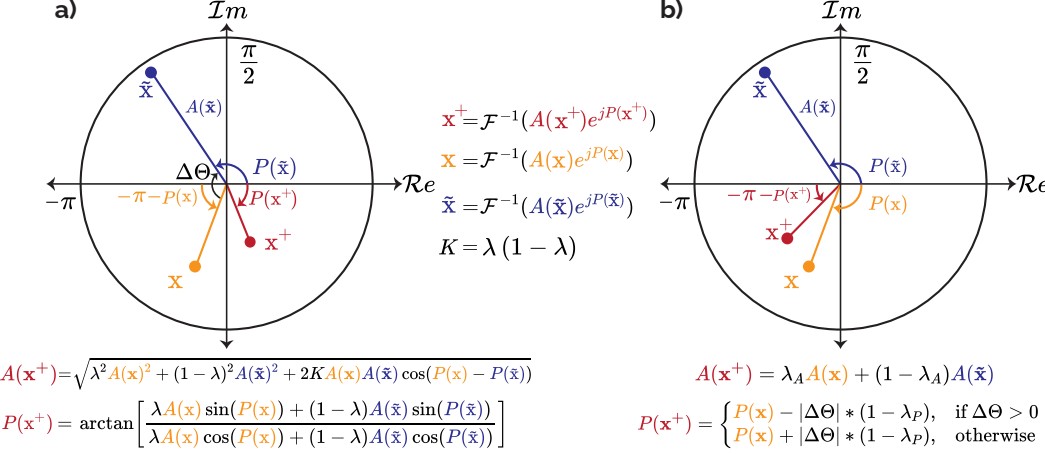

Figure 1: The phasor representation of linear mixup **a)**, and proposed mixup **b)**. The anchor, randomly chosen sample, and generated instances are represented as $\mathbf{x}$, $\tilde{\mathbf{x}}$, and $\mathbf{x}^+$, respectively.

**Theorem 3.1** (Guarantees for Mixing). *Under assumptions 2.1 and 2.2, given any $\lambda \in (0,1]$, the mutual information for the augmented instance lower bounded by the sampled $\lambda$ and anchor $\mathbf{x}$.*

$$\lambda \mathcal{I}(\mathbf{y};\mathbf{x}) \leq \mathcal{I}(\mathbf{y};\mathbf{x}^+) < \mathcal{I}(\mathbf{y};\mathbf{x}^*) \quad \text{where} \ \ \mathbf{x}^+ = \mathcal{F}^{-1}(A(\mathbf{x}^+)\angle P(\mathbf{x}^+)) \tag{5}$$

We provide an intuitive demonstration in Figure 1, along with a detailed mathematical proof presented in Appendix A. Our approach also offers increased flexibility in selecting the mixing coefficients of phase and magnitude, based on their sensitivities to the mixing process as well as the augmentation degree for each randomly chosen pair. Since the degree of augmentations has crucial importance for contrastive learning, there can be cases where augmentations are either too weak (intra-class features cannot be clustered together) or too strong (inter-class features can also collapse to the same point) and lead to sub-optimal results [17]. To mitigate this issue and find an order for augmentation degree, we search pairs of samples that are semantically closer, meaning they are more likely to belong to the same class. We then perform the proposed mixup more aggressively on these pairs, creating more closer and diverse samples while decreasing the augmentation strength for less similar pairs. To find similar samples without labels, we train a completely unsupervised $\beta$-VAE [32] that maps data points to a latent space such that two random samples are semantically similar if they are close in the latent as shown in Proposition 3.2.

**Proposition 3.2** (Consistency in Latent Space [33]). *Given a well-trained unconditional VAE with the encoder $E(.)$ that produces distribution $p_E(\mathbf{z}|\mathbf{x})$, the decoder $D(.)$ that produces distribution $q_D(\mathbf{x}|\mathbf{z})$ while the prior for $\mathbf{z}$ is $p(\mathbf{z})$, let $\mathbf{z_1}$ and $\mathbf{z_2}$ be two latent vectors of two different real samples $\mathbf{x_1}$ and $\mathbf{x_2}$, i.e., $E(\mathbf{x_1}) = \mathbf{z_1}$ and $E(\mathbf{x_2}) = \mathbf{z_2}$. if the distance $d(\mathbf{z_1},\mathbf{z_2}) \leq \delta$, then $D(\mathbf{z_1})$ and $D(\mathbf{z_2})$ will have a similar semantic label as in Equation 6.*

$$|\mathcal{I}(D(\mathbf{z_1});\mathbf{y}) - \mathcal{I}(D(\mathbf{z_2});\mathbf{y})| \leq \epsilon, \tag{6}$$

*where $\epsilon$ stands for tolerable semantic difference, $\delta$ is the maximum distance to maintain semantic consistency, and $d(.)$ is a distance measure such as cosine similarity between two vectors.*

The above proposition with Theorem 3.1 motivates us to perform augmentation aggressively if two randomly chosen samples are semantically closer. Therefore, we sample the mixup coefficient for both phase and magnitude from a uniform distribution $\lambda_A, \lambda_P \sim U(\beta, 1.0)$ with low values of $\beta$ if the distance between the latent vectors is below a threshold, otherwise, they are drawn from a truncated normal distribution $\lambda_A, \lambda_P \sim \mathcal{N}^t(\mu, \sigma, 1.0)$ with a high mean and low standard deviation.

## 4    Experiments

We conduct experiments on the proposed approach and compare it with other mixup methods or optimal/hard positive sample generation in the contrastive learning setup. During our experiments, we use SimCLR [15] framework without specialized architectures or a memory bank for all baselines to have a fair comparison. Results with other CL frameworks can be found in Appendix E. Complete training details and hyper-parameter settings for datasets and baselines are provided in Appendix D.

### 4.1    Datasets

We performed extensive experiments on eight datasets from three tasks that include activity recognition from inertial measurements (IMUs), heart rate prediction from photoplethysmography (PPG), and cardiovascular disease classification from electrocardiogram (ECG). We provided short descriptions of each dataset below, and further detailed information with metrics can be found in Appendix B.

***Activity recognition***    We used UCIHAR [34], HHAR [35], and USC [36] for activity recognition. During the evaluation, we assess the cross-person generalization capability of the contrastive models, i.e., the model is evaluated on a previously unseen target domain. We follow the settings in GILE [37] to treat each person as a single domain while the fine-tuning dataset is much smaller than the unsupervised one.

***Heart rate prediction***    We used the IEEE Signal Processing Cup in 2015 (IEEE SPC) [38], and DaLia [39] for PPG-based heart rate prediction. The SPC provides two datasets, one smaller with lesser artifacts (referred to as SPC12) [38] and a bigger dataset with more participants including heavy motions (referred to as SPC22). In line with previous studies, we adopted the leave-one-session-out (LOSO) cross-validation, which involves evaluating methods on subjects or sessions that were not used for pre-training and fine-tuning.

***Cardiovascular disease (CVD) classification***    We conducted our experiments on China Physiological Signal Challenge 2018 (CPSC2018) [40] and Chapman University, Shaoxing People's Hospital (Chapman) ECG dataset [41]. We selected the same four specific leads as in [42] while treating each dataset as a single domain with a small portion of the remainder dataset used for fine-tuning the pre-trained model. We split the dataset for fine-tuning and testing based on patients (each patient's recordings appear in only one set).

### 4.2    Baselines

**Comparison with prior mixup techniques**    We evaluate the effectiveness of our proposed mixup by comparing it with other commonly used mixup methods, including Linear-Mixup [31], Binary-Mixup [43], Geometric-Mixup [22], Cut-Mix [44], Amplitude-Mix [45] and Spec-Mix [46]. When we compare the performance of mixup techniques, we follow the same framework with [47] where the samples of mixture operation only happen in current batch samples. And, the mixup samples are paired with anchors, i.e., without applying mixup second times, for contrastive pre-training.

**Comparison with methods for optimal sample generation**    We evaluate the performance of our proposed method by comparing it with other data generation methods and baselines in contrastive learning setup while considering previously known augmentation techniques. Traditional data augmentations for time-series [19], such as resampling, flipping, adding noise, etc. InfoMin which leverages an adversarial training strategy to decrease the mutual information between samples

while maximizing the NCE loss [48]. NNCLR [49], which uses nearest neighbors in the learned representation space as the positive samples. Positive feature extrapolation [50], which creates hard positives through feature extrapolation. GenRep which uses the latent space of a generative model to generate "views" of the same semantic content by sampling nearby latent vectors [51]. Aug. Bank [21], which proposes an augmentation bank that manipulates frequency components of a sample with a limited budget. STAug [52], which combines spectral and time augmentation for generating samples using the empirical mode decomposition and linear mixup. DACL [22], which creates positive samples by mixing hidden representations. IDAA [53], which is an adversarial method by modifying the data to be hard positives without distorting the key information about their original identities using a VAE. More implementation details for each baseline are given in Appendix C.

### 4.3 Implementation

We use a combination of convolutional with LSTM-based network, which shows superior performance in many time-series tasks [19, 54, 55], as backbones for the encoder $f_\theta(.)$ where the projector is two fully-connected layers. We use InfoNCE as the loss, which is optimized using Adam with a learning rate of 0.003. We train with a batch size of 256 for 120 epochs and decay the learning rate using the cosine decay schedule. After pre-training, we train a single linear layer classifier on features extracted from the frozen pre-trained network, i.e., linear evaluation, with the same hyperparameters. Reported results are mean and standard deviation values across three independent runs with different seeds on the same split. More details about the implementation, architectures, and hyperparameters with the trained VAEs are given in Appendix D.

## 5 Results and Discussion

Tables 1, 2, and 3 present the results of our proposed approach compared to state-of-the-art methods for optimal/hard positive sample generation in contrastive learning setup across the three tasks in eight datasets. Additionally, Figure 2 compares our approach with prior mixup methods (e.g., linear mixup, cutmix) without applying any other additional augmentation techniques. Overall, our proposed method has demonstrated superior performance compared to other methods in seven datasets, with the second-best performance in the remaining dataset, and a minor performance gap.

Table 1: Performance Comparison of ours with prior works in *Activity Recognition* datasets

| Method | UCIHAR | | HHAR | | USC | |
|---|---|---|---|---|---|---|
| | ACC↑ | MF1↑ | ACC↑ | MF1↑ | ACC↑ | MF1↑ |
| *Supervised* | | | | | | |
| DCL [37] | 77.63 | – | 51.27 | – | 60.35 | – |
| CoDATS [56] | 68.22 | – | 45.69 | – | – | – |
| GILE [37] | 88.17 | – | 55.61 | – | – | – |
| *Self-Supervised* | | | | | | |
| Traditional Augs. | 87.05 ± 1.07 | 86.13 ± 0.96 | 85.48 ± 1.16 | 84.31 ± 1.31 | 53.47 ± 1.10 | 52.09 ± 0.95 |
| NNCLR [49] | 85.31 ± 0.91 | 83.56 ± 1.25 | 83.16 ± 1.32 | 82.15 ± 1.25 | 55.41 ± 1.43 | 52.64 ± 1.37 |
| InfoMin [48] | 38.07 ± 8.15 | 30.66 ± 9.15 | 31.58 ± 10.2 | 29.72 ± 11.1 | 35.89 ± 14.3 | 37.77 ± 9.12 |
| IDAA [53] | 82.23 ± 0.69 | 79.84 ± 0.89 | **88.98** ± 0.62 | **89.01** ± 0.55 | 59.23 ± 1.10 | 56.11 ± 1.54 |
| PosET [50] | 88.13 ± 0.91 | 87.35 ± 0.96 | 85.77 ± 1.11 | 85.90 ± 1.20 | 41.37 ± 5.63 | 39.43 ± 5.72 |
| STAug [52] | 89.83 ± 0.71 | 88.91 ± 0.62 | 87.69 ± 1.03 | 87.73 ± 0.93 | 55.61 ± 1.08 | 56.74 ± 1.21 |
| Aug. Bank [21] | 65.27 ± 1.12 | 71.16 ± 1.24 | 67.95 ± 1.45 | 75.13 ± 1.32 | 43.28 ± 4.37 | 47.31 ± 4.68 |
| GenRep [51] | 87.22 ± 1.05 | 86.48 ± 0.95 | 87.05 ± 0.95 | 86.45 ± 0.90 | 50.13 ± 2.85 | 49.50 ± 2.73 |
| DACL [22] | 73.12 ± 1.23 | 66.28 ± 1.11 | 80.89 ± 0.91 | 81.31 ± 0.78 | 53.61 ± 2.60 | 51.76 ± 2.21 |
| Ours | **91.60** ± 0.65 | **90.46** ± 0.53 | 88.05 ± 1.05 | 87.95 ± 1.10 | **60.13** ± 0.75 | **59.13** ± 0.69 |

From these tables, we can see that our proposed method significantly outperforms DACL, which suggests creating a positive sample by mixing fixed hidden representations in an intermediate layer [22], by a large margin (up to 20.8% with a 10.1% on average in activity recognition). This suggests that when the representations are not yet linearly separable at the beginning of the contrastive training process, the interpolated representations using mixup may be dissimilar to the actual interpolated samples and may not capture their underlying features. One interesting result from our experiments is that IDAA [53] exhibits comparable performance to our method in some datasets, and even slightly outperforms our approach in the HHAR dataset for activity recognition. Despite using distinct methods to generate positive instances, i.e., adversarial and mixup, our approach and IDAA algorithm

Table 2: Performance comparison of ours with prior works in *Heart Rate Prediction* datasets

| Method | IEEE SPC12 | | IEEE SPC 22 | | DaLia | |
|---|---|---|---|---|---|---|
| | MAE↓ | RMSE↓ | MAE↓ | RMSE↓ | MAE↓ | RMSE↓ |
| *Supervised* | | | | | | |
| DCL | 22.02 | 28.44 | 28.10 | 32.45 | 6.58 | 11.30 |
| CNN Ensemble* [39] | 3.89 | – | 8.74 | – | 8.58 | – |
| *Self-Supervised* | | | | | | |
| Traditional Augs. | $20.67 \pm 1.13$ | $26.35 \pm 0.98$ | $16.84 \pm 1.10$ | $22.23 \pm 0.72$ | $12.01 \pm 0.65$ | $21.09 \pm 0.86$ |
| NNCLR [49] | $20.28 \pm 2.21$ | $28.23 \pm 1.63$ | $23.49 \pm 1.54$ | $28.75 \pm 3.66$ | $11.56 \pm 0.63$ | $19.95 \pm 0.89$ |
| InfoMin [48] | $36.84 \pm 5.11$ | $29.78 \pm 7.31$ | $31.58 \pm 4.72$ | $29.72 \pm 4.83$ | $45.89 \pm 8.71$ | $50.77 \pm 9.72$ |
| IDAA [53] | $19.02 \pm 0.96$ | $27.42 \pm 1.11$ | $15.37 \pm 1.21$ | $22.41 \pm 1.42$ | $11.12 \pm 0.64$ | $20.45 \pm 0.69$ |
| PosET [50] | $25.60 \pm 1.93$ | $33.80 \pm 2.71$ | $23.42 \pm 1.50$ | $31.51 \pm 3.71$ | $35.99 \pm 3.95$ | $39.92 \pm 3.12$ |
| STAug [52] | $27.44 \pm 1.93$ | $35.63 \pm 3.10$ | $19.86 \pm 2.11$ | $30.70 \pm 3.54$ | $18.70 \pm 4.06$ | $30.81 \pm 3.61$ |
| Aug. Bank [21] | $27.31 \pm 2.17$ | $37.93 \pm 2.96$ | $27.84 \pm 2.03$ | $36.41 \pm 3.98$ | $35.87 \pm 4.18$ | $40.61 \pm 3.74$ |
| GenRep [51] | $21.02 \pm 1.41$ | $28.42 \pm 1.65$ | $15.67 \pm 1.23$ | $22.33 \pm 1.43$ | $25.41 \pm 1.62$ | $36.83 \pm 1.87$ |
| DACL [22] | $21.85 \pm 1.63$ | $28.17 \pm 1.75$ | $14.67 \pm 1.10$ | $20.06 \pm 1.21$ | $18.44 \pm 1.32$ | $25.61 \pm 1.45$ |
| Ours | $\mathbf{16.26} \pm 0.72$ | $\mathbf{22.48} \pm 0.95$ | $\mathbf{12.25} \pm 0.47$ | $\mathbf{18.20} \pm 0.61$ | $\mathbf{10.57} \pm 0.55$ | $\mathbf{20.37} \pm 0.73$ |

* The entire dataset, excluding the test, is utilized with labels, while in DCL, the labeled data size matches that of the CL

share similarities in approaches for positive instance generation in CL setup. The IDAA algorithm aims to create hard positive samples that lie near class boundaries without changing the identity of the sample, while our method interpolates two samples to produce a positive instance that is similar to the original sample while adding noise to the phase and amplitude in the direction of a randomly chosen sample. In other words, both approaches try to keep the sample identity intact by taking special precautions while generating new positive instances, which may explain their similar performance in our experiments.

In contrast, approaches that do not prioritize preserving sample identity while generating samples or hidden representations often demonstrate suboptimal performance on average while exhibiting increased variability across tasks.

Examples of such methods include PosET [50], which generates hard positive samples to improve contrastive learning by extrapolating features, STAug [52], which uses empirical mode decomposition with linear mixup technique together, and InfoMin [48], which tries to minimize mutual information between two instances in an adversarial manner. The performance comparison of prior mixup techniques and our proposed one is shown in Figure 2. On average, our proposed method outperforms all other mixup techniques while reducing the variance across tasks. What is interesting about this figure is that while the linear [31] and amplitude mixup [45] reach our method in some datasets for *activity recognition*, the performance of the linear mixup decreases heavily for the other two tasks whereas the amplitude mixup gives reasonable per-

Table 3: Performance comparison between ours and prior work in *CVD*.

| Method | CPSC 2018 | Chapman |
|---|---|---|
| | AUC↑ | AUC↑ |
| *Supervised* | | |
| CNN [57] | — | 95.80 |
| Casual CNN [58] | — | 97.70 |
| *Self-Supervised* | | |
| Traditional Augs. | $67.86 \pm 3.41$ | $74.69 \pm 2.04$ |
| NNCLR [49] | $70.06 \pm 2.05$ | $77.19 \pm 2.41$ |
| InfoMin [48] | $64.48 \pm 6.15$ | $56.34 \pm 9.12$ |
| IDAA [53] | $80.90 \pm 0.73$ | $93.63 \pm 0.91$ |
| PosET [50] | $72.58 \pm 2.12$ | $78.27 \pm 2.34$ |
| STAug [52] | $74.15 \pm 1.15$ | $93.88 \pm 0.87$ |
| Aug. Bank [21] | $81.78 \pm 1.24$ | $94.75 \pm 0.90$ |
| GenRep [51] | $52.49 \pm 3.43$ | $86.72 \pm 1.13$ |
| DACL [22] | $82.38 \pm 0.84$ | $92.28 \pm 0.97$ |
| Ours | $\mathbf{85.30} \pm 0.45$ | $\mathbf{95.90} \pm 0.82$ |

formance. This empirical outcome supports our initial theorem about the destructive effect of mixup, which suggests linear mixup or other derivatives can discard the whole task-specific information in the generated positive sample for quasi-periodic signals even though the mixing coefficient is sampled from a distribution such that the generated samples are much closer to the anchor.

## 5.1 Ablation Studies

Here, we present a comprehensive examination of our proposed method and the effect of its components on the performance. Mainly, we investigate the effect of the proposed mixup by applying the instance selection algorithm to the linear mixup (w/o Prop. Mixup). Then, we perform our proposed mixup with the constant $\lambda_A$ and $\lambda_P$ coefficients without investigating latent space distances between pairs (w/o Aug. Degree). Tables 4, 5 and 6 summarize the results. The second row in the tables shows the performance when the proposed mixup method is not applied while choosing mixup coefficients according to the distances in the latent space for the linear mixup. The last row illustrates

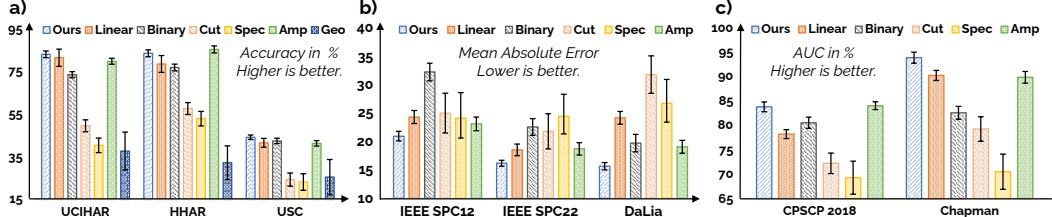

Figure 2: The comparison of mixup methods where the error bars represent the deviation across random seeds (explicit numbers are given in Appendix E). **a)** shows the performance in *activity recognition*, **b)** is for *heart rate prediction*, and finally **c)** shows the *CVD classification*. For the last two tasks, we excluded Geomix as its performance is extremely poor and distorts the y-axis scale.

the performance change resulting from randomly sampling mixup coefficients without considering any relationship between the selected pair while applying tailored mixup for phase and magnitude.

Table 4: Ablation on proposed mixup with coefficient selection in *Activity Recognition* datasets

| Method | UCIHAR | | HHAR | | USC | |
|---|---|---|---|---|---|---|
| | ACC↑ | MF1↑ | ACC↑ | MF1↑ | ACC↑ | MF1↑ |
| Ours | **91.60** ± 0.65 | **90.46** ± 0.53 | **88.05** ± 1.05 | **87.95** ± 1.10 | **60.13** ± 0.75 | **59.13** ± 0.69 |
| w/o Prop. Mixup | 83.09 (-8.51) | 81.65 (-8.81) | 85.89 (-2.16) | 86.01 (-1.94) | 45.10 (-15.03) | 43.64 (-15.49) |
| w/o Aug. Degree | 80.86 (-10.74) | 80.18 (-10.26) | 87.53 (-0.95) | 87.75 (-0.20) | 57.00 (-3.13) | 54.75 (-4.38) |

The results obtained from the ablation study support the previous claims and outcomes. For example, when the linear mixup is applied instead of the proposed mixup technique for *heart rate prediction* (Table 5, w/o Prop. Mixup), the performance decrease is significant compared to the case when coefficients are sampled without considering the distance in the latent space (Table 5, w/o Aug. Degree). This observation indicates that as the periodicity in data increases, linear mixup can lead to significant destructive interferences, whereas our method effectively prevents such issues.

Table 5: Ablation on proposed mixup with coefficient selection in *Heart Rate Prediction* datasets.

| Method | IEEE SPC12 | | IEEE SPC 22 | | DaLia | |
|---|---|---|---|---|---|---|
| | MAE↓ | RMSE↓ | MAE↓ | RMSE↓ | MAE↓ | RMSE↓ |
| Ours | **16.26** ± 0.72 | **22.48** ± 0.95 | **12.25** ± 0.47 | **18.20** ± 0.61 | **10.57** ± 0.55 | **20.37** ± 0.73 |
| w/o Prop. Mixup | 20.45 (+4.19) | 28.51 (+6.03) | 15.29 (+3.04) | 24.08 (+5.88) | 24.11 (+13.54) | 35.45 (+15.18) |
| w/o Aug. Degree | 19.30 (+3.04) | 24.84 (+2.36) | 16.01 (+3.76) | 21.21 (+3.19) | 11.10 (+0.53) | 20.13 (-0.24) |

While our mixup technique consistently enhances performance across datasets, we observe a decline when the mixing coefficients are sampled based on the distance in the latent space for two datasets. Also, the performance increase gained by sampling coefficients based on distance is relatively low compared to the proposed mixup. Several factors can explain this observation. First, the VAE might not be

Table 6: Ablation on proposed mixup with coefficient selection in *CVD*.

| Method | CPSC 2018 | Chapman |
|---|---|---|
| | AUC↑ | AUC↑ |
| Ours | **85.30** ± 0.45 | **95.90** ± 0.82 |
| w/o Prop. Mixup | 81.20 (-4.10) | 86.30 (-9.60) |
| w/o Aug. Degree | 80.67 (-4.63) | 95.98 (+0.08) |

well trained due to the limited size of data in each class, i.e., the assumption in Proposition 3.2 does not hold. This can lead to inconsistencies in the semantic similarity of the latent space such that two close samples in the latent space might have different labels. Second, if the number of classes increases for a downstream task, the probability of sampling intra-class samples in a batch will decrease, leading to a lack of performance improvement. Therefore, in future investigations, it might be beneficial to use a different distance metric for quasi-periodic time-series data such that it can scale with the number of classes while considering the lack of big datasets.

More ablation studies about the sensitivity of mixing coefficients and performance in different self-supervised learning frameworks, like BYOL [59] can be found in Appendix E.1 and E.2. And, investigations regarding whether we still need known data augmentations are given in Appendix E.3. Examples that visually demonstrate the negative effects of linear mixup and our proposed mixup

technique to prevent this problem can be found in Appendix F. Comparative results regarding the performance of the tailored mixup in the supervised learning paradigm are given in Appendix G.

## 6 Related Work

The goal of contrastive learning is to contrast positive with negative pairs [60]. In other words, the embedding space is governed by two forces, the attraction of positive pairs and the repellence of negatives, actualized through the contrastive loss [61]. Since label information is unavailable during the training, positive pairs are generated using augmentation techniques on a single sample, while negative pairs are randomly sampled from the entire dataset. Therefore, the choice or generation of positive and negative samples plays a pivotal role in the success of contrastive learning [62–65] and both approaches, generation/selection of positive/negative pairs, have been investigated thoroughly in the literature [66–71], we limit our discussion about prior works related to data augmentation techniques that create optimal or hard samples without labels.

**Adversarial based approaches**   A growing body of literature has investigated generating samples by using adversarial training for both positives and negatives [48, 72, 73]. A seminal work about the importance of augmentations in CL, InfoMin, presented an adversarial training strategy where players try to minimize and maximize the mutual information using the NCE loss [48]. CLAE, one of the first works that leveraged the adversarial approach, shows that adversarial training generates challenging positive and hard negative pairs [72]. Another recent study proposed an adversarial approach that generates hard samples while retaining the original sample identity by leveraging the identity-disentangled feature of VAEs [53]. However, adversarial augmentations may change the original sample identity due to excessive perturbations and it is infeasible to tune the attack strength for every sample to preserve the identity. In other words, these approaches do not consider the sample-specific features and use a constant perturbation coefficient for all samples whereas our proposed method considers the similarity between pairs and tunes the mixing coefficients accordingly.

**Mixup based approaches**   Mixup-based methods have been recently explored in contrastive learning [22, 71, 47, 74, 75]. According to a recent theoretical work [22], mixup has implicit data-adaptive regularization effects that promote generalization better than adding Gaussian noise, which is a commonly used augmentation strategy in both time-series and vision data [76–78]. Although, mixup-based approaches have shown success in different problems [79, 80], such as domain adaptation, creating samples using mixup in the input space is infeasible in domains where data has a non-fixed topology, such as sequences, trees, and graphs [22]. Therefore, recent works suggest mixing hidden representations of samples, similar to Manifold Mixup [81]. However, this method claims that mixing fixed-length hidden representation via an intermediate layer "$\mathbf{z} = \alpha \mathbf{z} + (1 - \alpha)\bar{\mathbf{z}}$" can be interpreted as adding noise to a given sample in the direction of another. However, it is an overly optimistic claim because early during training, where in most cases there is usually no linear separability among the representations, this synthesis may result in hidden representations that are completely different and far away from the samples [71, 82]. Therefore, in this work, we take a different approach and modify the mixup method considering its limitations for quasi-periodic non-stationary time-series data. Also, unlike most existing methods that aim to generate hard samples—samples that are close to class boundaries—using adversarial approaches [53, 48, 72] or feature extrapolation [50, 71], our method seeks to connect semantically closer samples together using interpolation in a tailored way.

## 7 Conclusion

In this paper, we first demonstrate the destructive effect of linear mixup for quasi-periodic time-series data, then introduce a novel tailored mixup method to generate positive samples for the contrastive learning formulation while preventing this destructive effect and interpolating the samples appropriately. Theoretically, we show that our proposed method guarantees the interpolation of pairs without causing any loss of information while generating a diverse set of samples. Empirically, our method outperforms the prior approaches in three real-world tasks. By conducting experiments on contrastive and supervised learning settings, we show that our approach is agnostic to the choice of learning paradigm. Thus, it holds the potential for utilization in generating augmented data for different learning paradigms as well. We believe that the method proposed in this paper has the potential to significantly improve learning solutions for a diverse range of time series tasks.

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

# Appendix

## A   Proof

In this section, we present complete proofs of our theoretical study, starting with notations.

### A.1   Notations

Fourier transform of a real-valued sample with a finite duration is obtained as in Equation 7.

$$X_k = \mathcal{F}(\mathbf{x}) = \sum_{n=-\infty}^{\infty} \mathbf{x}_n e^{-j2\pi kn} \tag{7}$$

The amplitude and phase for each frequency are calculated from the Fourier transform as follows.

$$\begin{aligned} A(\mathbf{x}) &= \sqrt{\mathrm{Re}(X_k)^2 + \mathrm{Im}(X_k)^2} \\ P(\mathbf{x}) &= \mathrm{arctan2}(\mathrm{Im}(X_k), \mathrm{Re}(X_k)), \end{aligned} \tag{8}$$

where arctan is a 2-argument arctangent which is the angle measure in radians. The phasor, as in Figure 1, of a sample is represented as in Equation 9.

$$X_k = \mathcal{F}(\mathbf{x}) = A(\mathbf{x})e^{jP(\mathbf{x})} \tag{9}$$

### A.2   Proof for Proposition 2.3

**Proposition A.1** (Destructive Mixup). *If Assumptions 2.1 and 2.2 hold, there exist $\lambda \sim Beta(\alpha, \alpha)$ or $\lambda \sim U(\beta, 1.0)$ with high values of $\beta$ such that when linear mixup techniques are utilized, the lower bound of the mutual information for the augmented sample distribution decreases to zero.*

$$\begin{aligned} 0 \leq \mathcal{I}(\mathbf{y}; \mathbf{x}^+) &< \mathcal{I}(\mathbf{y}; \mathbf{x}^*) \quad \text{where} \\ \mathbf{x}^+ = \lambda\mathbf{x} + (1-\lambda)\tilde{\mathbf{x}} \;\; \text{and} & \int_{f^*} S_{x^*}(f) = \int_{-\infty}^{\infty} S_{x^*}(f) \end{aligned} \tag{10}$$

*Proof.*

$$\mathbf{x}^+ = \lambda\mathbf{x} + (1-\lambda)\tilde{\mathbf{x}} \tag{11}$$

From the linearity of Fourier transformation and ignoring k in $X_k$ for the sake of easiness.

$$X^+ = \lambda\mathbf{X} + (1-\lambda)\tilde{\mathbf{X}} \tag{12}$$

$$X^+ = \tilde{\mathbf{X}} + \lambda(\mathbf{X} - \tilde{\mathbf{X}}) \tag{13}$$

Let $\tilde{\mathbf{X}} = e^{-j\omega\phi_k}\mathbf{X}\boldsymbol{\omega}_k$, where $\phi_k$ and $\boldsymbol{\omega}_k$ are random phase and frequency modulators for each frequency, sampled from distributions $\phi_k \sim \Phi$, $\boldsymbol{\omega}_k \sim \Omega$.

$$X^+ = e^{-j\omega\phi_k}\mathbf{X}\boldsymbol{\omega}_k + \lambda(\mathbf{X} - e^{-j\omega\phi_k}\mathbf{X}\boldsymbol{\omega}_k) \tag{14}$$

$$X^+ = \mathbf{X}\left[\lambda + e^{-j\omega\phi_k}\boldsymbol{\omega}_k - \lambda e^{-j\omega\phi_k}\boldsymbol{\omega}_k\right] \tag{15}$$

$$X^+ = \mathbf{X}\left[\lambda + (1-\lambda)e^{-j\omega\phi_k}\boldsymbol{\omega}_k\right] \tag{16}$$

$$X^+ = \mathbf{X}\left[\lambda + (1-\lambda)\boldsymbol{\omega}_k(\cos(\omega\phi_k) - j\sin(\omega\phi_k))\right] \tag{17}$$

From the quasi-periodicity, assume that the frequency ranges of interest ($f^*$, i.e., $k^*$) are overlapped for both samples while the sampled random modulators have the following relationship.

$$\boldsymbol{\omega}_{k^*} \approx \frac{\lambda}{1 - \lambda} \ \text{ and } \ \theta \equiv [\omega\boldsymbol{\phi}_{k^*}] \ (\text{mod } 2\pi), \tag{18}$$

where $\theta$ is an odd multiple of $\pi$. Equation 17 can be simplified as follows.

$$\mathrm{X}_{k^*}^+ = \mathbf{X}_{k^*} \left[\lambda + \lambda\cos\left(\omega\boldsymbol{\phi}_{k^*}\right)\right] \tag{19}$$

$$\mathrm{X}_{k^*}^+ \approx 0 \tag{20}$$

$$\mathrm{X}_{k^*}^+ = \sum_{n=-\infty}^{\infty} \mathbf{x}_n e^{-j2\pi k^* n} \longrightarrow \sum_{n=-\infty}^{\infty} \mathbf{x}_n e^{-j2\pi k^* n} \approx 0 \tag{21}$$

$$S_{x^+}(f^*) = \lim_{N\to\infty} \frac{1}{2N} \left| \sum_{n=-N}^{N} x_n e^{-j2\pi f^* n} \right|^2 \tag{22}$$

From Assumption 2.1,

$$\mathcal{I}(\mathbf{y}; \mathbf{x}^+) \propto \int_{f^*} S_{x^+}(f) \, / \int_{-\infty}^{\infty} S_{x^+}(f) \tag{23}$$

$$0 \leq \mathcal{I}(\mathbf{y}; \mathbf{x}^+) < \mathcal{I}(\mathbf{y}; \mathbf{x}^*) \tag{24}$$

We use Euler's formula to expand Equation 16 to 17. While we use the frequency bins ($k$) and frequency values in Hz ($f$) interchangeably for Equations 21 and 22. $\qquad\square$

Although the above proof is to show the resulting instances may not contain any task-relevant information, it can also be demonstrated that the augmentation process can potentially discard the partial task-specific information (not whole) if $\boldsymbol{\phi}_k$ and $\boldsymbol{\omega}_k$ are close to indicated relationships.

## A.3 Proof for Theorem 3.1

**Theorem A.2** (Guarantees for Mixing). *Under assumptions 2.1 and 2.2, given any $\lambda \in (0, 1]$, the mutual information for the augmented instance lower bounded by the sampled $\lambda$ and anchor $\mathbf{x}$.*

$$\lambda \mathcal{I}(\mathbf{y}; \mathbf{x}) \leq \mathcal{I}(\mathbf{y}; \mathbf{x}^+) < \mathcal{I}(\mathbf{y}; \mathbf{x}^*) \quad \text{where} \quad \mathbf{x}^+ = \mathcal{F}^{-1}(A(\mathbf{x}^+) \angle P(\mathbf{x}^+)) \tag{25}$$

*Proof.*

$$\mathbf{x}^+ = \mathcal{F}^{-1}(A(\mathbf{x}^+) \angle P(\mathbf{x}^+)) \quad \text{where}$$
$$A(\mathbf{x}^+) = \lambda_A A(\mathbf{x}) + (1 - \lambda_A) A(\tilde{\mathbf{x}}) \quad \text{and}$$
$$P(\mathbf{x}^+) = \begin{cases} P(\mathbf{x}) - |\Delta\Theta| * (1 - \lambda_P), & \text{if } \Delta\Theta > 0 \\ P(\mathbf{x}) + |\Delta\Theta| * (1 - \lambda_P), & \text{otherwise} \end{cases} \tag{26}$$

$$\mathrm{X}^+ = A(\mathbf{x}^+) e^{jP(\mathbf{x}^+)} \tag{27}$$

$$\left| \mathrm{X}^+ \right| = \left| A(\mathbf{x}^+) e^{jP(\mathbf{x}^+)} \right| \tag{28}$$

$$\left| \mathrm{X}^+ \right| = A(\mathbf{x}^+) \quad \text{where} \quad \left| \mathrm{X}_k^+ \right| = \left| \sum_{n=-\infty}^{\infty} \mathbf{x}_n^+ e^{-j2\pi kn} \right| \tag{29}$$

$$\left| \mathrm{X}^+ \right| = \lambda A(\mathbf{x}) + (1 - \lambda) A(\tilde{\mathbf{x}}) \tag{30}$$

$$\left| \mathrm{X}^+ \right| = \lambda \left| \sum_{n=-\infty}^{\infty} \mathbf{x}_n e^{-j2\pi kn} \right| + (1 - \lambda) \left| \sum_{n=-\infty}^{\infty} \tilde{\mathbf{x}}_n e^{-j2\pi kn} \right| \tag{31}$$

$$\lambda \left| \sum_{n=-\infty}^{\infty} \mathbf{x}_n e^{-j2\pi kn} \right| + (1 - \lambda) \left| \sum_{n=-\infty}^{\infty} \tilde{\mathbf{x}}_n e^{-j2\pi kn} \right| \geq \lambda \left| \sum_{n=-\infty}^{\infty} \mathbf{x}_n e^{-j2\pi kn} \right| \tag{32}$$

$$\int_{f^*} S_{x^+}(f) \geq \lambda \int_{f^*} S_x(f) \tag{33}$$

Using the $\int_{-\infty}^{\infty} S_{x^+}(f) = \int_{\infty}^{\infty} S_{\tilde{x}}(f)$ (i.e., both samples are normalized to have the same power) and Assumption 2.1,

$$\mathcal{I}(\mathbf{y}; \mathbf{x}^+) \propto \int_{f^*} S_{x^+}(f) \, / \int_{-\infty}^{\infty} S_{x^+}(f) \tag{34}$$

$$\lambda \mathcal{I}(\mathbf{y}; \mathbf{x}) \leq \mathcal{I}(\mathbf{y}; \mathbf{x}^+) < \mathcal{I}(\mathbf{y}; \mathbf{x}^*) \tag{35}$$

Proof is completed with Equation 35 by combining equations 33 and 34. $\qquad\square$

Although this proof ignores the effect of phase mixing on the mutual information with the assumption 2.1, it is known that phase components carry semantically important features [45]. Therefore, it is necessary to note that the objective of this proof is to demonstrate that by applying mixup separately to the phase and amplitude components, we can avoid destructive interference.

# B  Datasets

In this section, we give details about the datasets that are used during our experiments.

## B.1  Human Activity Recognition

**UCIHAR**  Human activity recognition using smartphones dataset (UCIHAR) [34] is collected by 30 subjects within an age range of 16 to 48 performing six daily living activities with a waist-mounted smartphone. Six activities include walking, sitting, lying, standing, walking upstairs, and walking downstairs. Data is captured by 3-axial linear acceleration and 3-axial angular velocity at a constant rate of 50 Hz. We used the pre-processing technique the same as in [37, 19] such that the input contains nine channels with 128 features (it is sampled in sliding window of 2.56 seconds and 50% overlap, resulting in 128 features for each window). Windows are normalized to zero mean and unit standard deviation before feeding to models. Also, we follow the same experimental setup with prior works as follows. The experiments are conducted with a leave-one-domain-out strategy, where one of the domains is chosen to be the unseen target [19]. The contrastive pre-training is conducted with all subjects without any label information except the target one. Training of the linear layer, which is added to the frozen trained encoder, is only performed with the first five subjects of UCIHAR after excluding the target subject. In other words, if the target subject is 0, the subjects from 1 to 29 are used to train the encoder without any label information. Then, subjects from 1 to 4 are used to train the linear layer. And, evaluation is performed for subject 0. This is performed for the first five subjects with three random seeds and the mean value is reported.

**HHAR**  Heterogeneity Dataset for Human Activity Recognition (HHAR) is collected by nine subjects within an age range of 25 to 30 performing six daily living activities with eight different smartphones—Although HHAR includes data from smartwatches as well, we use data from smartphones—that were kept in a tight pouch and carried by the users around their waists [35]. Subjects then perform 6 activities: 'bike', 'sit', 'stairs down', 'stairs up', 'stand', and 'walk'. Due to variant sampling frequencies of smart devices used in HHAR dataset, we downsample the readings to 50 Hz and apply 100 (two seconds) and 50 as sliding window length with step size, the windows are normalized to zero mean with unit standard deviation. We used the first four subjects (i.e., a, b, c, d) as source domains.

**USC**  USC human activity dataset (USC-HAD) is composed of 14 subjects (7 male, 7 female, aged 21 to 49 with a mean of 30.1) executing 12 activities with a sensor on the front right hip. The data dimension is six (3-axis accelerometer, 3-axis gyroscope) and the sample rate is 100 Hz. 12 activities include walking forward, walking left, walking right, walking upstairs, walking downstairs, running forward, jumping up, sitting, standing, sleeping, elevator up, and elevator down. We used the pre-processing technique with a smaller window size such that the input contains six channels with 100 features (it is sampled in a sliding window of 1 second and 50% overlap, resulting in 100 features for each window). The same normalization is also applied to windows before feeding to models. We used the same setup with UCIHAR while source subjects are chosen as the last four this time.

## B.2  Heart Rate Prediction

**IEEE SPC**  This competition provided a training dataset of 12 subjects (SPC12) and a test dataset of 10 subjects [39]. The IEEE SPC dataset overall has 22 recordings of 22 subjects, ages ranging from 18 to 58 performing three different activities [83]. Each recording has sampled data from three accelerometer signals and two PPG signals along with the sampled ECG data and the sampling frequency is 125 Hz. All these recordings were recorded from the wearable device placed on the wrist of each individual. All recordings were captured with a 2-channel pulse oximeter with green LEDs, a tri-axial accelerometer, and a chest ECG for the ground-truth HR estimation. During our experiments, we used PPG channels. We choose the first five subjects of SPC12 as source domains similar to *activity recognition* setup while the last six subjects of SPC22 are used for source domains to prevent overlapping subjects with SPC12.

**Dalia**  PPG dataset for motion compensation and heart rate estimation in Daily Life Activities (DaLia) was recorded from 15 subjects (8 females, 7 males, mean age of 30.6), where each recording was approximately two hours long. PPG signals were recorded while subjects went through different

daily life activities, for instance sitting, walking, driving, cycling, working, and so on. PPG signals were recorded at a sampling rate of 64 Hz. The first five subjects are used as source domains.

All PPG datasets are standardized as follows. Initially, a fourth-order Butterworth bandpass filter with a frequency range of 0.5–4 Hz is applied to PPG signals. Subsequently, a sliding window of 8 seconds with 2-second shifts is employed for segmentation, followed by z-score normalization of each segment. Lastly, the signal is resampled to a frequency of 25 Hz for each segment.

### B.3 Cardiovascular disease (CVD) classification

**CPSC** China Physiological Signal Challenge 2018 (CPSC2018), held during the 7th International Conference on Biomedical Engineering and Biotechnology in Nanjing, China. This dataset consists of 6,877 (male: 3,699; female: 3,178) and 12 lead ECG recordings lasting from 6 seconds to 60 seconds with 500 Hz. We use the original labelling [40] with one normal and eight abnormal types as follows: atrial fibrillation, first-degree atrioventricular block, left bundle branch block, right bundle branch block, premature atrial contraction, premature ventricular contraction, ST-segment depression, ST-segment elevated. We resampled recordings to 100 Hz and excluded recordings of less than 10 seconds.

**Chapman** Chapman University, Shaoxing People's Hospital (Chapman) ECG dataset which provides 12-lead ECG with 10 seconds of a sampling rate of 500 Hz. The recordings are downsampled to 100 Hz, resulting in each ECG frame consisting of 1000 samples. The labeling setup follows the same approach as in [41] with four classes: atrial fibrillation, GSVT, sudden bradycardia, and sinus rhythm. The ECG frames are normalized to have a mean of 0 and scaled to have a standard deviation of 1. We split the dataset to 80–20% for training and testing as suggested in [41].

We choose leads I, II, III, and V2 during our experiments for both ECG datasets. We followed a similar setup with prior works [57] and considered each dataset as a single domain different from previous tasks. The fine-tuning of the linear layer, which is added to the frozen pre-trained encoder, is performed with 80% of the same domain.

### B.4 Metrics

We used the common evaluation metrics in the literature for each task. Specifically, we used accuracy (Acc) and F1 score for *activity recognition* [19], mean absolute error (MAE), and root mean square error (RMSE) for *heart rate prediction* [39, 84], and the area under the ROC curve (AUC) for *cardiovascular disease classification* [57].

In this section, we explain how to calculate each metric for different time-series tasks. For activity recognition, the accuracy metric is computed by dividing the sum of true positives and true negatives by the total number of samples where a window has a single label. The MF1 score is calculated as a harmonic mean of the precision and recall where metrics are obtained globally by counting the total true positives, false negatives, and false positives similar to [19].

For heart rate prediction, the Mean Absolute Error (MAE) and Root-Mean-Square Error (RMSE) are calculated using the following equation:

$$\text{MAE} = \frac{1}{K}\sum_{k=1}^{K}|\text{HR}model(k) - \text{HR}ref(k)| \tag{36}$$

$$\text{RMSE} = \sqrt{\frac{\sum_{k=1}^{K}(\text{HR}model(k) - \text{HR}ref(k))^2}{K}}, \tag{37}$$

where K represents the total number of segments. The variables $\text{HR}model(k)$ and $\text{HR}ref(k)$ denote the output of the model and reference heart rate values in beats-per-minute for the $k^{th}$ segment, respectively. This performance metric is commonly used in PPG-based heart rate estimation studies [39]. The estimated heart rate values ($\text{HR}model(k)$) are obtained using our model, while the reference heart rate values ($\text{HR}ref(k)$) are directly taken from datasets.

The AUC score for CVD classification is calculated using the one-vs-one scheme where the average AUC is computed for all possible pairwise combinations of classes for both datasets.

## C  Baselines

### C.1  Prior Mixup Techniques

In this section, we give a detailed explanation of each mixup technique we compare our proposed method.

**LinearMix**  We apply linear mixup as in Equation 38 to generate positive samples, if $\mathbf{x}$ has more than one channel, mixup is applied independently for each of them.

$$\mathbf{x}^+ = \lambda \mathbf{x} + (1 - \lambda)\tilde{\mathbf{x}} \tag{38}$$

**BinaryMix**  We implement the binary mixup [43] by swapping the elements of $\mathbf{x}$ with the elements of another randomly chosen sample $\tilde{\mathbf{x}}$ as shown below.

$$\mathbf{x}^+ = \mathbf{m} \odot \mathbf{x} + (1 - \mathbf{m}) \odot \tilde{\mathbf{x}}, \tag{39}$$

where $\mathbf{m}$ is a binary mask sampled from a Bernouilli($\rho$) with high values, and $\odot$ stands for Hadamard product.

**GeometricMix**  In Geometric Mixup, we create a positive sample corresponding to a sample $\mathbf{x}$ by taking its weighted-geometric mean with another randomly chosen sample $\tilde{\mathbf{x}}$ same as [22] as shown below.

$$\mathbf{x}^+ = \mathbf{x}^\lambda + \tilde{\mathbf{x}}^{(1-\lambda)} \tag{40}$$

**CutMix**  Cutmix is implemented similarly to Binarymix. However, instead of changing each sample point with a probability, we cut a continuous portion using a rectangle mask $\mathbf{M}$ from a signal $\mathbf{x}$ and replace it with the same portion of another randomly chosen one $\tilde{\mathbf{x}}$. The starting point of the mask is uniformly sampled while its length is sampled from lower values such that the augmented sample is more similar to the anchor. If the signal has multiple channels, this process is applied to all channels in the same section.

$$\mathbf{x}^+ = \mathbf{M} \odot \mathbf{x} + (1 - \mathbf{M}) \odot \tilde{\mathbf{x}}, \quad \text{and} \quad \mathbf{M} = \text{rect}\left(\frac{b}{a}\right), \tag{41}$$

where $b$ and $a$ are the starting point and length of the rectangle wave, respectively.

**AmplitudeMix**  AmplitudeMix is introduced for domain adaptation problems by mixing the amplitude information of images without mixing the phase of two samples [45]. In our setup, we perform amplitude mixing on the time series data across all channels while keeping the phase component unchanged. In other words, we perform the following operations.

$$\mathbf{x}^+ = \mathcal{F}^{-1}(A(\mathbf{x}^+)\angle P(\mathbf{x}^+)) \quad \text{where}$$
$$A(\mathbf{x}^+) = \lambda_A A(\mathbf{x}) + (1 - \lambda_A)A(\tilde{\mathbf{x}}) \quad \text{and} \quad P(\mathbf{x}^+) = P(\mathbf{x}) \tag{42}$$

**SpecMix**  We implement the SpecMix by applying CutMix to the spectrogram of time-series where the spectrogram is calculated using the short-time Fourier transform as follows.

$$\mathrm{X}_k^+ = \sum_{n=-\infty}^{\infty} \mathbf{x}_n g[n - mR]e^{-j2\pi kn}, \tag{43}$$

where $g[n - mR]$ is an analysis window of length M with hop length of R over the signal and calculating the discrete Fourier transform (DFT) of each segment of windowed data. The length of the Fourier transform is set to the sample size of the input time series while the hop and window parameters are set to the quarter of the length.

## C.2  Prior Methods for Sample Generation

In this section, we give a detailed explanation of prior methods for data generation methods.

**Traditional Augmentations**   We apply two separate data augmentation to the anchor for creating two instances, and the encoders are trained to maximize agreement using the contrastive loss in [15]. We search mainly for augmentations that are known in state-of-the-art works [19]. The detailed augmentations are given in Table 22.

**InfoMin**   We train a model $g_\theta(.)$, which is restricted to sample-wise $1 \times 1$ convolutions and ReLU activations same as in [48], to decrease the mutual information between two instances. In the original paper, the input sample is split into two instances ($X_1$ and $X_{2:3}$) and then adversarial training is performed. As we do not have RGB channels for time-series data, we added Gaussian noise to the signal for creating other instances and then perform adversarial training.

**NNCLR**   We follow a similar setup to SimCLR by applying two separate data augmentations, then we use nearest neighbors in the learned representation space as the positive in contrastive losses [49].

**PosET**   We perform the dimension level mixing with extrapolation of positive features as follows:

$$\mathbf{z}^+ = \lambda \odot \mathbf{z} + (1 - \lambda) \odot \tilde{\mathbf{z}}, \tag{44}$$

where $\odot$ is Hadamard product, and $\lambda \sim Beta(\alpha, \alpha)$. We add 1 to sampled $\lambda$ for extrapolation as in [50].

**GenRep**   In the original implementation of GenRep, the authors use implicit generative models (IGMs) such as BigBiGAN [85] that are trained with millions of images to create the anchor and positive instance by sampling nearby latent vectors. However, as the number of samples for training is limited in time series and there is a well-trained generator for different time-series tasks, we use our trained VAE for sampling nearby latent vectors as positives. Mainly, we sample an anchor from real data, feed it to the encoder, add a Gaussian noise sampled from a truncated normal distribution, and use the output of the decoder for the positive sample with the anchor.

**STAug**   The Spectral and Time Augmentation (STAug) method is specifically proposed for the time-series forecasting task, where the authors apply the empirical mode decomposition to decompose time series into multiple subcomponents, then reassemble these subcomponents with random weights to generate new synthetic series. Finally, in the time domain, the method uses the linear mixup to generate samples from the reassembled components. Although, the mixing coefficient sampled from a beta distribution in the original implementation, we observe significant performance decreases when the same distribution with parameters is used in our experiments, possibly due to the generated samples being far away from the anchor. We, therefore, investigate the case when the mixing coefficient is sampled from uniform distribution with high values, e.g., same as our method. Since there is no

**Augmentation Bank**   The augmentation bank that perturbs frequency components of a time-series signal is proposed in [21] where the authors use it for unsupervised domain adaptation with a different framework than SimCLR, namely time-frequency consistency (TF-C). As it is a novel data augmentation technique, we have implemented the frequency augmentation bank as a baseline while using the SimCLR framework for a fair comparison with other methods. The authors also employed a collection of time-based augmentations for the time-domain contrastive encoder. Nonetheless, since these augmentations have already been studied in previous CL setups, we chose to exclusively utilize the frequency augmentation bank. In the paper, the authors mentioned using a small budget with low-frequency perturbations results in a performance increase, thus we chose the budget with a single frequency while choosing the $\alpha = 0.5$ with the same settings in the paper.

**DACL** We perform the mixup for hidden representations, i.e., before applying projection-head, as follows.

$$\mathbf{v}^+ = \lambda\mathbf{v} + (1 - \lambda)\tilde{\mathbf{v}}, \tag{45}$$

where $\mathbf{v}$ is the fixed-length hidden representations of samples while $\lambda$ is sampled from uniform distribution with high values.

**IDAA** We follow the original implementation of authors with their proposed VAE architecture while optimizing the adversarial strength for each time-series task. We apply the FGSM adversarial attack the same as in the original implementation [53] by perturbing the encoded representation of a sample while adding noises along the gradient sign's direction of the loss.

One setup difference between this section and the previous mixup methods is that when we compare our work with PosET, GenRep, DACL, and IDAA, we apply the best traditional data augmentation techniques, which are used for SimCLR implementation, to the specific positive data generation mechanisms. The reason for this approach is that the original implementations of certain works indicate that the proposed methods achieve optimal results when used in conjunction with known augmentations, where our observations align with these findings.

The detailed hyperparameters for each baseline with the corresponding time series tasks are given in the following section.

# D   Implementation Details

## D.1   Parameters for mixing

In this section, we provide the parameters that are used during our experiments. To determine the optimal parameters of the baselines for each task, we conduct a grid search. This search is performed on a small validation set taken from the largest dataset of the respective tasks, which are USC, Dalia and Chapman. We believe that this approach ensures fairness and produces more realistic results, as dataset-specific optimizations can lead to overfitting of parameters, particularly in smaller and less diverse datasets.

Table 7: Parameters for baselines

| Method | *Activity Recognition* | *Heart rate Prediction* | *CVD Classification* |
|---|---|---|---|
| Linear Mixup | $\lambda \sim U(0.9, 1)$ | $\lambda \sim U(0.9, 1)$ | $\lambda \sim U(0.85, 1)$ |
| Binary Mixup | $m \sim U(0.8, 1)$ | $m \sim U(0.9, 1)$ | $m \sim U(0.9, 1)$ |
| Geometric Mixup | $\lambda \sim U(0.9, 1)$ | $\lambda \sim U(0.9, 1)$ | $\lambda \sim U(0.9, 1)$ |
| CutMix | $b \sim U(0, 1)$ $a \sim U(0.1, 0.4)$ | $b \sim U(0, 1)$ $a \sim U(0.1, 0.3)$ | $b \sim U(0, 1)$ $a \sim U(0.1, 0.3)$ |
| AmplitudeMix | $\lambda_A \sim U(0.9, 1)$ | $\lambda_A \sim U(0.9, 1)$ | $\lambda_A \sim U(0.8, 1)$ |
| SpecMix | $b \sim U(0, 1)$ $a \sim U(0.1, 0.4)$ | $b \sim U(0, 1)$ $a \sim U(0.1, 0.3)$ | $b \sim U(0, 1)$ $a \sim U(0.1, 0.3)$ |
| PosET | $\lambda \sim Beta(2, 2)$ | $\lambda \sim Beta(2, 2)$ | $\lambda \sim Beta(2, 2)$ |
| GenRep | $\lambda \sim \mathcal{N}^t(0, 0.2, 1.0)$ | $\lambda \sim \mathcal{N}^t(0, 0.25, 1.0)$ | $\lambda \sim \mathcal{N}^t(0, 0.2, 1.0)$ |
| DACL | $\lambda \sim U(0.9, 1)$ | $\lambda \sim U(0.9, 1)$ | $\lambda \sim U(0.85, 1)$ |
| IDAA | $\delta = 0.1$ | $\delta = 0.15$ | $\delta = 0.2$ |
| Ours | $\lambda_A \sim U(0.7, 1), \lambda_P \sim U(0.9, 1)$ $\epsilon = 0.7, \lambda_A, \lambda_P \sim \mathcal{N}^t(0.9, 0.1, 0.9)$ | $\lambda_A \sim U(0.7, 1), \lambda_P \sim U(0.9, 1)$ $\epsilon = 0.8, \lambda_A, \lambda_P \sim \mathcal{N}^t(1, 0.1, 0.9)$ | $\lambda_A \sim U(0.7, 1), \lambda_P \sim U(0.9, 1)$ $\epsilon = 0.7, \lambda_A, \lambda_P \sim \mathcal{N}^t(1, 0.1, 0.9)$ |

## D.2   Baseline Encoder Architecture

For the baseline encoder model, we adopt the DeepConvLSTM as in [19] where the architecture has 4 convolutional layers with $5 \times 1$ size of 64 kernels while ReLU is followed each convolution. After the convolutions, the tensor is passed through a dropout layer with a dropout rate of 0.5 to prevent overfitting. Then, the output of dropout is fed into the 2-layer LSTM with 128 units. After training

the baseline encoder, we attach a linear layer and freeze the previous layers for fine-tuning. This architecture is widely used for the datasets we used during our experiments [37, 83, 19], we therefore adopt the same network across tasks.

## D.3    VAE Models

We use the total correlation variational autoencoder ($\beta$-TCVAE) [86] to calculate the distance between two encoded samples in the latent space. We train the model for 100 epochs with a learning rate of $1e-3$ while setting the batch size to 2048. The latent dimensions and the $\beta$ values are set to 10 and 5, respectively. Below, we present the tables providing detailed information about the architectures of the encoder and decoder for datasets. The output of convolutional layers is fed to the batch normalization before the activation layer is applied. For tasks *Heart rate Prediction* and *CVD Classification*, we use task-specific encoder and decoder as the number of channels and input size for datasets in each task are the same. However, two different networks, one for UCIHAR and one for others, are designed for the *Activity Recognition* due to different number of input channels.

Table 8: Encoder Network for UCIHAR in *Activity Recognition*

| Encoder | | | | | |
|---|---|---|---|---|---|
| Layer Name | Output size | # of kernels | Kernel size | Stride | Activation |
| Input | Nx1x128x9 | | | | |
| Convolution | Nx32x60x7 | 32 | 9x3 | 2x1 | ReLU |
| Convolution | Nx32x27x5 | 32 | 7x3 | 2x1 | ReLU |
| Convolution | Nx64x8x3 | 64 | 5x3 | 3x1 | ReLU |
| Convolution | Nx128x2x1 | 128 | 5x3 | 2x1 | ReLU |
| Convolution | Nx512x1x1 | 512 | 2x1 | 1x1 | ReLU |
| Convolution | Nx20x1x1 | 10 | 1x1 | 1x1 | |

Table 9: Decoder Network for UCIHAR in *Activity Recognition*

| Decoder | | | | | |
|---|---|---|---|---|---|
| Layer Name | Output size | # of kernels | Kernel size | Stride | Activation |
| Input | Nx1x10x1 | | | | |
| Transposed Convolution | Nx512x2x9 | 512 | 2x9 | 1x1 | ReLU |
| Transposed Convolution | Nx128x8x9 | 128 | 4x1 | 6x1 | ReLU |
| Transposed Convolution | Nx64x16x9 | 64 | 4x1 | 2x1 | ReLU |
| Transposed Convolution | Nx32x32x9 | 32 | 4x1 | 2x1 | ReLU |
| Transposed Convolution | Nx32x64x9 | 32 | 4x1 | 2x1 | ReLU |
| Transposed Convolution | Nx1x128x9 | 1 | 4x1 | 2x1 | |

Table 10: Encoder Network for USC and HHAR in *Activity Recognition*

| Encoder | | | | | |
|---|---|---|---|---|---|
| Layer Name | Output size | # of kernels | Kernel size | Stride | Activation |
| Input | Nx1x100x6 | | | | |
| Convolution | Nx32x46x5 | 32 | 9x2 | 2x1 | ReLU |
| Convolution | Nx32x20x4 | 32 | 9x2 | 2x1 | ReLU |
| Convolution | Nx64x8x3 | 64 | 5x2 | 2x1 | ReLU |
| Convolution | Nx128x2x2 | 128 | 5x2 | 2x1 | ReLU |
| Convolution | Nx512x1x1 | 512 | 2x2 | 1x1 | ReLU |
| Convolution | Nx20x1x1 | 10 | 1x1 | 1x1 | |

Table 11: Decoder Network for USC and HHAR in *Activity Recognition*

| | Decoder | | | | |
|---|---|---|---|---|---|
| Layer Name | Output size | # of kernels | Kernel size | Stride | Activation |
| Input | Nx1x10x1 | | | | |
| Transposed Convolution | Nx512x2x6 | 512 | 2x6 | 1x1 | ReLU |
| Transposed Convolution | Nx128x6x6 | 128 | 6x1 | 2x1 | ReLU |
| Transposed Convolution | Nx64x12x6 | 64 | 4x1 | 2x1 | ReLU |
| Transposed Convolution | Nx32x25x6 | 32 | 5x1 | 2x1 | ReLU |
| Transposed Convolution | Nx32x50x6 | 32 | 4x1 | 2x1 | ReLU |
| Transposed Convolution | Nx1x100x6 | 1 | 4x1 | 2x1 | |

Table 12: Encoder Network for *Heart rate Prediction*

| | Encoder | | | | |
|---|---|---|---|---|---|
| Layer Name | Output size | # of kernels | Kernel size | Stride | Activation |
| Input | Nx1x200x1 | | | | |
| Convolution | Nx32x94x1 | 32 | 13x1 | 2x1 | ReLU |
| Convolution | Nx32x43x1 | 32 | 9x1 | 2x1 | ReLU |
| Convolution | Nx64x18x1 | 64 | 9x1 | 2x1 | ReLU |
| Convolution | Nx128x6x1 | 128 | 7x1 | 2x1 | ReLU |
| Convolution | Nx512x1x1 | 512 | 5x1 | 2x1 | ReLU |
| Convolution | Nx20x1x1 | 20 | 2x1 | 1x1 | |

Table 13: Decoder Network for *Heart rate Prediction*

| | Decoder | | | | |
|---|---|---|---|---|---|
| Layer Name | Output size | # of kernels | Kernel size | Stride | Activation |
| Input | Nx1x10x1 | | | | |
| Transposed Convolution | Nx512x6x1 | 512 | 6x1 | 1x1 | ReLU |
| Transposed Convolution | Nx128x12x1 | 128 | 4x1 | 2x1 | ReLU |
| Transposed Convolution | Nx64x25x1 | 64 | 5x1 | 2x1 | ReLU |
| Transposed Convolution | Nx32x50x1 | 32 | 4x1 | 2x1 | ReLU |
| Transposed Convolution | Nx32x100x1 | 32 | 4x1 | 2x1 | ReLU |
| Transposed Convolution | Nx1x200x1 | 1 | 4x1 | 2x1 | |

Table 14: Encoder Network for *CVD Classification*

| | Encoder | | | | |
|---|---|---|---|---|---|
| Layer Name | Output size | # of kernels | Kernel size | Stride | Activation |
| Input | Nx1x1000x4 | | | | |
| Convolution | Nx32x330x3 | 32 | 12x2 | 3x1 | ReLU |
| Convolution | Nx32x107x2 | 32 | 10x2 | 3x1 | ReLU |
| Convolution | Nx64x34x1 | 64 | 8x2 | 3x1 | ReLU |
| Convolution | Nx128x9x1 | 128 | 8x1 | 3x1 | ReLU |
| Convolution | Nx512x1x1 | 512 | 7x1 | 3x1 | ReLU |
| Convolution | Nx20x1x1 | 20 | 1x1 | 1x1 | |

Table 15: Decoder Network for *CVD Classification*

| | Decoder | | | | |
|---|---|---|---|---|---|
| Layer Name | Output size | # of kernels | Kernel size | Stride | Activation |
| Input | Nx1x10x1 | | | | |
| Transposed Convolution | Nx512x4x4 | 512 | 6x1 | 1x1 | ReLU |
| Transposed Convolution | Nx128x12x4 | 128 | 4x1 | 2x1 | ReLU |
| Transposed Convolution | Nx64x36x4 | 64 | 5x1 | 2x1 | ReLU |
| Transposed Convolution | Nx32x109x4 | 32 | 4x1 | 2x1 | ReLU |
| Transposed Convolution | Nx32x331x4 | 32 | 4x1 | 2x1 | ReLU |
| Transposed Convolution | Nx1x1000x4 | 1 | 4x1 | 2x1 | |

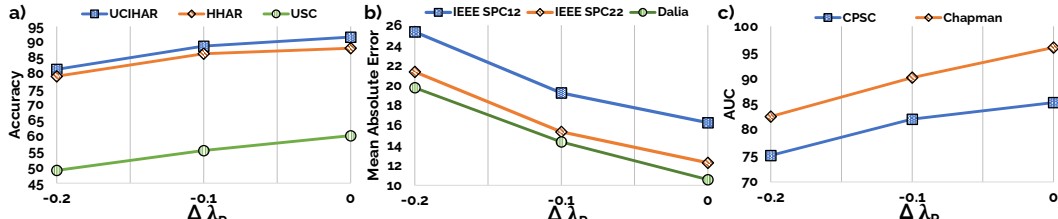

Figure 3: The experiment regarding the effect of phase mixup coefficients in eight datasets. **a)** shows the performance in *activity recognition*, **b)** is for *heart rate prediction* using PPG, and finally **c)** shows the *cardiovascular disease classification*

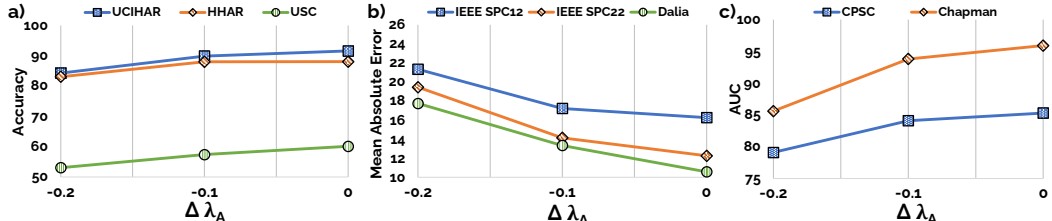

Figure 4: The experiment regarding the effect of amplitude mixup coefficients in eight datasets. **a)** shows the performance in *activity recognition*, **b)** is for *heart rate prediction* using PPG, and finally **c)** shows the *cardiovascular disease classification*

# E    Additonal Results

## E.1    The effect and robustness of mixing coefficients

In this section, our experiments focus on observing the impact of a diverse range of mixing coefficients for both phase and amplitude components. We decrease the lower threshold of distributions for sampling the mixing coefficient by $0.1$ and $0.2$. For example, normally the phase mixup coefficient for *Activity Recognition* is sampled from truncated normal $\lambda_P \sim \mathcal{N}^t(1, 0.1, 0.9)$ and uniform $\lambda_P \sim U(0.9, 1)$. We decrease the low threshold value from $0.9$ to $0.8$ and $0.7$ and report the results for both phase and amplitude. The results are reported in Figures 3 and 4 for eight datasets.

From Figures 3 and 4, it can be inferred that the phase component is more sensitive to the changes. In other words, a significant decrease in performance is observed when the mixing coefficients for the phase are sampled from lower values whereas this effect is not as much as severe for the amplitude coefficient, indicating that the amplitude of frequencies is more robust to changes compared to phase.

## E.2    The performance in other frameworks

In this section, we investigate the effect of data augmentations in three different unsupervised learning frameworks which are SimCLR [15], BYOL [59] and TS-TCC [16]. For BYOL, the hidden size of the projector is set to 128, the exponential moving average parameter is set to 0.996. For TS-TCC, the $\lambda_1$ and $\lambda_2$ coefficients of temporal and contextual contrasting losses are set to 1, the same as in the original implementation. In TS-TCC, the authors proposed to use a weak (jitter and scale) and strong (permutation and jitter) augmentation together, which is shown as TS-TCC + Traditional Augs in the tables. During our experiments, we followed the original implementation of TS-TCC with their proposed architecture and applied additional augmentations after the strong one without changing the original contrastive learning framework. We set the scaling ratio to 2 and 10 for permutation (splitting the signal into a random number of segments with a maximum of 10 and randomly shuffling them). These parameters for augmentation strengths are set to the same values as in the original implementation.

Table 16: Performance comparison of our method in different CL frameworks for *Activity Recognition*

| Method | UCIHAR | | HHAR | | USC | |
|---|---|---|---|---|---|---|
| | ACC↑ | MF1↑ | ACC↑ | MF1↑ | ACC↑ | MF1↑ |
| SimCLR + Traditional Augs. | 87.05 ± 1.07 | 86.13 ± 0.96 | 85.48 ± 1.16 | 84.31 ± 1.31 | 53.47 ± 1.10 | 52.09 ± 0.95 |
| SimCLR + Aug. Bank | 65.27 ± 1.12 | 71.16 ± 1.24 | 67.95 ± 1.45 | 75.13 ± 1.32 | 43.28 ± 4.37 | 47.31 ± 4.68 |
| SimCLR + DACL | 73.12 ± 1.23 | 66.28 ± 1.11 | 80.89 ± 0.91 | 81.31 ± 0.78 | 53.61 ± 2.60 | 51.76 ± 2.21 |
| SimCLR + Ours | 91.60 ± 0.65 | 90.46 ± 0.53 | 88.05 ± 1.05 | 87.95 ± 1.10 | **60.13** ± 0.75 | **59.13** ± 0.69 |
| BYOL + Traditional Augs. | 83.41 ± 0.95 | 82.13 ± 1.12 | 86.41 ± 0.97 | 86.31 ± 1.10 | 58.34 ± 1.15 | 55.04 ± 1.15 |
| BYOL + Aug. Bank | 73.71 ± 0.74 | 69.80 ± 1.10 | 84.60 ± 0.93 | 84.65 ± 1.03 | 52.00 ± 1.21 | 49.14 ± 1.18 |
| BYOL + DACL | 73.86 ± 1.12 | 70.46 ± 1.24 | 82.76 ± 1.04 | 84.89 ± 0.93 | 47.14 ± 2.08 | 45.34 ± 2.98 |
| BYOL + Ours | 87.01 ± 1.10 | 84.92 ± 1.13 | **90.31** ± 1.16 | **90.45** ± 1.31 | 56.87 ± 0.91 | 55.01 ± 0.95 |
| TS-TCC + Traditional Augs. | 90.95 ± 0.87 | 90.30 ± 0.64 | 35.57 ± 1.43 | 40.13 ± 1.67 | 39.76 ± 1.61 | 43.12 ± 1.10 |
| TS-TCC + Aug. Bank | 76.78 ± 0.95 | 76.52 ± 0.97 | 20.25 ± 1.54 | 19.25 ± 1.32 | 21.37 ± 1.78 | 20.15 ± 1.48 |
| TS-TCC + DACL | 73.86 ± 1.12 | 70.46 ± 1.24 | 33.89 ± 1.87 | 37.41 ± 1.39 | 36.74 ± 1.36 | 40.18 ± 1.45 |
| TS-TCC + Ours | **91.86** ± 0.97 | **91.92** ± 1.02 | 38.45 ± 1.12 | 43.52 ± 1.33 | 42.61 ± 1.92 | 45.06 ± 1.11 |

Table 17: Performance comparison of our method in different CL frameworks for *Heart Rate Prediction*

| Method | IEEE SPC12 | | IEEE SPC22 | | DaLia | |
|---|---|---|---|---|---|---|
| | MAE↓ | RMSE↓ | MAE↓ | RMSE↓ | MAE↓ | RMSE↓ |
| SimCLR + Traditional Augs. | 20.67 ± 1.13 | 26.35 ± 0.98 | 16.84 ± 1.10 | 22.23 ± 0.72 | 12.01 ± 0.65 | 21.09 ± 0.86 |
| SimCLR + Aug. Bank | 27.31 ± 2.17 | 37.93 ± 2.96 | 27.84 ± 2.03 | 36.41 ± 3.98 | 35.87 ± 4.18 | 40.61 ± 3.74 |
| SimCLR + DACL | 21.85 ± 1.63 | 28.17 ± 1.75 | 14.67 ± 1.10 | 20.06 ± 1.21 | 18.44 ± 1.32 | 25.61 ± 1.45 |
| SimCLR + Ours | 16.26 ± 0.72 | 22.48 ± 0.95 | **12.25** ± 0.47 | **18.20** ± 0.61 | **10.57** ± 0.55 | **20.37** ± 0.73 |
| BYOL + Traditional Augs. | 20.68 ± 0.98 | 27.11 ± 0.85 | 21.16 ± 1.10 | 26.83 ± 1.05 | 12.03 ± 0.75 | 20.77 ± 0.83 |
| BYOL + Aug. Bank | 26.08 ± 1.05 | 32.62 ± 0.93 | 21.87 ± 1.03 | 29.13 ± 1.03 | 18.63 ± 0.91 | 28.30 ± 0.87 |
| BYOL + DACL | 26.45 ± 1.23 | 33.50 ± 1.32 | 21.29 ± 1.13 | 27.34 ± 1.33 | 15.11 ± 0.93 | 23.21 ± 0.83 |
| BYOL + Ours | 19.85 ± 0.88 | 26.10 ± 0.94 | 22.08 ± 1.24 | 28.20 ± 1.13 | 11.45 ± 0.63 | 20.38 ± 0.80 |
| TS-TCC + Traditional Augs. | 11.08 ± 1.03 | 16.97 ± 0.92 | 16.10 ± 1.23 | 26.11 ± 1.11 | 16.18 ± 1.03 | 24.27 ± 0.95 |
| TS-TCC + Aug. Bank | 11.44 ± 1.01 | 17.06 ± 0.94 | 13.79 ± 1.21 | 22.41 ± 1.08 | 17.28 ± 1.12 | 25.41 ± 0.98 |
| TS-TCC + DACL | 11.60 ± 1.16 | 18.26 ± 1.20 | 15.25 ± 1.26 | 24.40 ± 1.10 | 16.27 ± 1.16 | 24.28 ± 0.97 |
| TS-TCC + Ours | **10.82** ± 0.65 | **16.93** ± 0.73 | 13.63 ± 1.02 | 21.80 ± 1.11 | 15.90 ± 0.57 | 23.81 ± 0.89 |

Tables 16 17 and 18 compares the performance of three data augmentation techniques, traditional time-series augmentations, DACL and our proposed method, in contrastive learning frameworks of BYOL, SimCLR, and TS-TCC.

Table 18: Performance comparison of our method in different CL frameworks for *CVD classification*

| Method | CPSC 2018 | Chapman |
|---|---|---|
| | AUC↑ | AUC↑ |
| SimCLR + Traditional Augs. | 67.86 ± 3.41 | 74.69 ± 2.04 |
| SimCLR + Aug. Bank | 81.78 ± 1.24 | 94.75 ± 0.90 |
| SimCLR + DACL | 82.38 ± 0.84 | 92.28 ± 0.97 |
| SimCLR + Ours | 85.30 ± 0.45 | **95.90** ± 0.82 |
| BYOL + Traditional Augs | 75.41 ± 1.34 | 85.63 ± 1.43 |
| BYOL + Aug. Bank | 83.51 ± 1.12 | 91.03 ± 1.18 |
| BYOL + DACL | 77.61 ± 1.16 | 81.62 ± 1.24 |
| BYOL + Ours | 83.25 ± 1.03 | 91.23 ± 1.15 |
| TS-TCC + Traditional Augs | 87.07 ± 1.10 | 92.03 ± 1.17 |
| TS-TCC + Aug. Bank | 86.67 ± 1.04 | 92.15 ± 1.02 |
| TS-TCC + DACL | 87.63 ± 0.83 | 92.21 ± 0.86 |
| TS-TCC + Ours | **88.05** ± 0.37 | 92.11 ± 0.75 |

The results show that the BYOL is more robust to the choice of augmentations than SimCLR, which is also indicated in the original paper [59]. Also, another important outcome of this ablation experiment is that when the TS-TCC framework is used for datasets HHAR and USC, the performance decreases compared to other datasets. A possible explanation for this decrease in the TS-TCC might be the hyper-parameters of the augmentations that are used in the paper. The authors change the strength of the permutation window from dataset to dataset. In our experiments, we used the same hyperparameter for all activity recognition datasets, which can explain the outcome. This ablation experiment also shows that the degree of traditional augmentations is important for contrastive learning to learn class invariant representations.

### E.3 Do we still need data augmentations?

In this section, we conduct experiments to observe the performance of methods without additional augmentations. During our experiments, we searched for the best traditional augmentation technique for each method in a given task. We searched over common time series augmentation methods in literature (Table 22), and applied them with baselines. Specifically, we apply *Resample* for *Activity Recognition*, *Permutation with Noise* for *Heart rate Prediction* and *Noise with Scaling* for *CVD Classification*. We have observed that these augmentations yield the best results for all baselines when applied prior to the proposed techniques. However, for GenRep, we found that applying the augmentations after generating instances results in better performance, similar to the original work [51]. We, therefore, apply these specified augmentations for each baseline and report the corresponding results.

Different from other baselines, we observed performance increases for a few datasets when GenRep is applied without any augmentations. This phenomenon can be attributed to the generation of low-quality and less realistic positive samples, where additional augmentations lead to alterations in semantic information, due to less number of samples during training VAE models. However, in the end, we observe that applying additional augmentations always increases the performance on average for all baselines in each task.

Table 19: Performance comparison of methods without Augs. in *Activity Recognition* datasets

| Method | UCIHAR | | HHAR | | USC | |
|---|---|---|---|---|---|---|
| | ACC↑ | MF1↑ | ACC↑ | MF1↑ | ACC↑ | MF1↑ |
| IDAA [53] | 82.23 ± 0.69 | 79.84 ± 0.89 | **88.98** ± 0.62 | **89.01** ± 0.55 | 59.23 ± 1.10 | 56.11 ± 1.54 |
| w/o Aug. | 64.42 (-17.81) | 65.17 (-14.67) | 86.44 (-2.54) | 86.31 (-2.70) | 35.22 (-24.01) | 33.62 (-22.59) |
| GenRep [51] | 87.22 ± 1.05 | 86.48 ± 0.95 | 87.05 ± 0.95 | 86.45 ± 0.90 | 50.13 ± 2.85 | 49.50 ± 2.73 |
| w/o Aug. | 88.01 (+0.79) | 88.12 (+1.64) | 86.51 (-0.54) | 86.33 (-0.22) | 48.31 (-1.82) | 47.33 (-2.17) |
| DACL [22] | 73.12 ± 1.23 | 66.28 ± 1.11 | 80.89 ± 0.91 | 81.31 ± 0.78 | 53.61 ± 2.60 | 51.76 ± 2.21 |
| w/o Aug. | 45.17 (-27.95) | 44.84 (-21.44) | 56.70 (-24.19) | 56.55 (-25.76) | 27.12 (-26.49) | 26.99 (-24.77) |
| Ours | **91.60** ± 0.65 | **90.46** ± 0.53 | 88.05 ± 1.05 | 87.95 ± 1.10 | **60.13** ± 0.75 | **59.13** ± 0.69 |
| w/o Aug. | 84.04 (-5.56) | 83.34 (-7.12) | 86.70 (-1.35) | 86.72 (-1.23) | 45.55 (-14.58) | 44.94 (-14.19) |

Table 20: Performance comparison of methods without Augs. in *Heart Rate Prediction* datasets

| Method | IEEE SPC12 | | IEEE SPC22 | | DaLia | |
|---|---|---|---|---|---|---|
| | MAE↓ | RMSE↓ | MAE↓ | RMSE↓ | MAE↓ | RMSE↓ |
| IDAA [53] | 19.02 ± 0.96 | 27.42 ± 1.11 | 15.37 ± 1.21 | 22.41 ± 1.42 | 11.12 ± 0.64 | 20.45 ± 0.69 |
| w/o Aug. | 20.19 (+1.17) | 28.51 (+1.09) | 16.34 (+0.97) | 25.75 (+3.34) | 16.01 (+4.89) | 25.62 (+5.17) |
| GenRep [51] | 21.02 ± 1.41 | 28.42 ± 1.65 | 15.67 ± 1.23 | 22.33 ± 1.43 | 25.41 ± 1.62 | 36.83 ± 1.87 |
| w/o Aug. | 20.51 (-0.51) | 28.35 (-0.07) | 23.07 (+7.40) | 33.20 (+10.87) | 20.03 (-5.38) | 31.01 (-5.82) |
| DACL [22] | 21.85 ± 1.63 | 28.17 ± 1.75 | 14.67 ± 1.10 | 20.06 ± 1.21 | 18.44 ± 1.32 | 25.61 ± 1.45 |
| w/o Aug. | 22.75 (+0.90) | 29.90 (+1.73) | 20.88 (+6.21) | 29.51 (+2.70) | 28.24 (+9.45) | 37.33 (+11.72) |
| Ours | **16.26** ± 0.72 | **22.48** ± 0.95 | **12.25** ± 0.47 | **18.20** ± 0.61 | **10.57** ± 0.55 | **20.37** ± 0.73 |
| w/o Aug. | 19.41 (+3.15) | 26.23 (+3.75) | 16.41 (+4.16) | 25.71 (+7.51) | 16.73 (+6.16) | 27.43 (+7.06) |

Table 21: Performance comparison of methods without Augs. in *CVD classification* datasets

| Method | CPSC 2018 AUC↑ | Chapman AUC↑ |
|---|---|---|
| IDAA [53] | $80.90 \pm 0.73$ | $93.63 \pm 0.91$ |
| w/o Aug. | 79.00 (-1.90) | 92.37 (-1.26) |
| GenRep [51] | $52.49 \pm 3.43$ | $86.72 \pm 1.13$ |
| w/o Aug. | 45.17 (-7.32) | 84.51 (-2.21) |
| DACL [22] | $82.38 \pm 0.84$ | $92.28 \pm 0.97$ |
| w/o Aug. | 73.00 (-9.38) | 75.10 (-17.18) |
| Ours | $\mathbf{85.30} \pm 0.45$ | $\mathbf{95.90} \pm 0.82$ |
| w/o Aug. | 79.67 (-5.63) | 93.48 (-2.42) |

Table 22: Common time series augmentations [19]

| Domain | Augmentation | Details |
|---|---|---|
| *Time* | Noise | Add Gaussian noise sampled from normal distribution, $\mathcal{N}(0, 0.4)$ |
| | Scale | Amplify channels by a random distortion sampled from normal distribution $\mathcal{N}(2, 1.1)$ |
| | Shuffle | Randomly permute the channels of the sample. (Not available for *Heart rate Prediction*) |
| | Negate | Multiply the value of the signal by a factor of -1 |
| | Permute | Split signals into no more than 5 segments, then permute the segments and combine them into the original shape |
| | Resample | Interpolate the time-series to 3 times its original sampling rate and randomly down-sample to its initial dimensions |
| | Rotation | Rotate the 3-axial (x, y, and z) readings of each IMU sensor by a random degree, which follows a uniform around a random axis in the 3D space. (Only applied for *Activity Recognition*) |
| | Time Flip | Flip the time series in time for all channels, i.e., $\mathbf{x}_{Aug}[n] = \mathbf{x}[-n]$ |
| | Random Zero Out | Randomly chose a section to zero out |
| | Permutation + Noise | Combination of Permutation and Noise |
| | Noise + Scale | Combination of Noise and Scaling |
| *Frequency* | Highpass | Apply a highpass filter in the frequency domain to reserve high-frequency components |
| | Lowpass | Apply a lowpass filter in the frequency domain to reserve low-frequency components |
| | Phase shift | Shift the phase of time-series data with a randomly generalized number |
| | Noise in Frequency | Add Gaussian noise, sampled from normal distribution $\mathcal{N}(0, 0.5)$, to the frequency spectrum |

## E.4 The effect of interpolating phase components

Here, we investigate the effect of phase interpolation of two samples on the CL performance. In our proposed method, we bring the phase components of the two coherent signals together by adding a small value to the anchor's phase in the direction of the other sample. In this section, we apply the opposite case of our proposed method and increase the gap of phase difference between the anchor and randomly chosen sample. However, we mix their amplitudes according to our proposed method to only observe the phase effect. In other words, we perform the mixup as in Equation 46. Note that the phase mixing in Equation 46 differs from the proposed method only by the sign change.

$$
\begin{aligned}
\mathbf{x}^+ &= \mathcal{F}^{-1}(A(\mathbf{x}^+)\angle P(\mathbf{x}^+)) \quad \text{where} \\
A(\mathbf{x}^+) &= \lambda_A A(\mathbf{x}) + (1 - \lambda_A)A(\tilde{\mathbf{x}}) \quad \text{and} \\
P(\mathbf{x}^+) &= P(\mathbf{x}) + \Delta\Theta * (1 - \lambda_P)
\end{aligned}
\tag{46}
$$

Also, It is important to note that we sample the mixing coefficients for both amplitude and phase from the same distributions in the proposed method to have a fair comparison. Tables 23 24 25.

Table 23: Performance comparison of our method and its ablation regarding the phase interpolation in SimCLR and BYOL frameworks for *Activity Recognition*

| Method | UCIHAR | | HHAR | | USC | |
|---|---|---|---|---|---|---|
| | ACC↑ | MF1↑ | ACC↑ | MF1↑ | ACC↑ | MF1↑ |
| SimCLR + Traditional Augs. | $87.05 \pm 1.07$ | $86.13 \pm 0.96$ | $85.48 \pm 1.16$ | $84.31 \pm 1.31$ | $53.47 \pm 1.10$ | $52.09 \pm 0.95$ |
| SimCLR + Phase Gap | $79.62 \pm 1.10$ | $80.57 \pm 1.03$ | $86.55 \pm 0.83$ | $86.68 \pm 0.71$ | $53.61 \pm 2.60$ | $51.76 \pm 2.21$ |
| SimCLR + Ours | $\mathbf{91.60} \pm 0.65$ | $\mathbf{90.46} \pm 0.53$ | $88.05 \pm 1.05$ | $87.95 \pm 1.10$ | $\mathbf{60.13} \pm 0.75$ | $\mathbf{59.13} \pm 0.69$ |
| BYOL + Traditional Augs. | $83.41 \pm 0.95$ | $82.13 \pm 1.12$ | $86.41 \pm 0.97$ | $86.31 \pm 1.10$ | $58.34 \pm 1.15$ | $55.04 \pm 1.15$ |
| BYOL + Phase Gap | $78.66 \pm 0.63$ | $75.45 \pm 1.02$ | $85.82 \pm 0.91$ | $85.16 \pm 0.92$ | $56.14 \pm 0.67$ | $56.20 \pm 0.75$ |
| BYOL + Ours | $87.01 \pm 1.10$ | $84.92 \pm 1.13$ | $\mathbf{90.31} \pm 1.16$ | $\mathbf{90.45} \pm 1.31$ | $56.87 \pm 0.91$ | $55.01 \pm 0.95$ |

Table 24: Performance comparison of our method and its ablation regarding the phase interpolation in SimCLR and BYOL frameworks for *Heart Rate Prediction*

| Method | IEEE SPC12 | | IEEE SPC22 | | DaLia | |
|---|---|---|---|---|---|---|
| | MAE↓ | RMSE↓ | MAE↓ | RMSE↓ | MAE↓ | RMSE↓ |
| SimCLR + Traditional Augs. | $20.67 \pm 1.13$ | $26.35 \pm 0.98$ | $16.84 \pm 1.10$ | $22.23 \pm 0.72$ | $12.01 \pm 0.65$ | $21.09 \pm 0.86$ |
| SimCLR + Phase Gap | $18.90 \pm 1.43$ | $25.29 \pm 1.56$ | $14.60 \pm 1.03$ | $19.84 \pm 1.15$ | $17.57 \pm 1.13$ | $27.72 \pm 1.35$ |
| SimCLR + Ours | $\mathbf{16.26} \pm 0.72$ | $\mathbf{22.48} \pm 0.95$ | $\mathbf{12.25} \pm 0.47$ | $\mathbf{18.20} \pm 0.61$ | $\mathbf{10.57} \pm 0.55$ | $\mathbf{20.37} \pm 0.73$ |
| BYOL + Traditional Augs. | $20.68 \pm 0.98$ | $27.11 \pm 0.85$ | $21.16 \pm 1.10$ | $26.83 \pm 1.05$ | $12.03 \pm 0.75$ | $20.77 \pm 0.83$ |
| BYOL + Phase Gap | $25.93 \pm 0.96$ | $32.68 \pm 0.90$ | $21.87 \pm 1.03$ | $29.13 \pm 1.03$ | $17.46 \pm 0.83$ | $27.24 \pm 0.83$ |
| BYOL + Ours | $19.85 \pm 0.88$ | $26.10 \pm 0.94$ | $22.08 \pm 1.24$ | $28.20 \pm 1.13$ | $11.45 \pm 0.63$ | $20.38 \pm 0.80$ |

Table 25: Performance comparison of our method and its ablation regarding the phase interpolation in SimCLR and BYOL frameworks for *CVD classification*

| Method | CPSC 2018 | Chapman |
|---|---|---|
| | AUC↑ | AUC↑ |
| SimCLR + Traditional Augs. | $67.86 \pm 3.41$ | $74.69 \pm 2.04$ |
| SimCLR + Phase Gap | $77.45 \pm 1.10$ | $91.95 \pm 0.91$ |
| SimCLR + Ours | $\mathbf{85.30} \pm 0.45$ | $\mathbf{95.90} \pm 0.82$ |
| BYOL + Traditional Augs | $75.41 \pm 1.34$ | $85.63 \pm 1.43$ |
| BYOL + Phase Gap | $83.11 \pm 1.03$ | $91.02 \pm 1.11$ |
| BYOL + Ours | $83.25 \pm 1.03$ | $91.23 \pm 1.15$ |

### E.5 The comparison of Mixup methods

In this section, we give a detailed comparison of prior mixup methods with ours below tables, which are the explicit numbers for Figure 2. Our method demonstrates superior performance compared to previous mixup techniques in 11 out of 14 metrics, indicating its effectiveness. Additionally, the Amplitude Mixup technique, which yields comparable results in two datasets, further supports our claim regarding the destructive effect of simultaneously mixing phase and magnitude for time series. The relatively lower performance of Amplitude Mixup for some datasets can be explained by its limited diversity in generating positive samples since this technique has no solution for mixing the phase of samples in randomly chosen pairs. In other words, as the phase of the augmented instance is the same as the anchor in Amplitude Mix, the diversity of generated positive samples is less compared to other techniques.

Table 26: Performance comparison of ours with prior mixups in *Activity Recognition* datasets

| Method | UCIHAR | | HHAR | | USC | |
|---|---|---|---|---|---|---|
| | ACC↑ | MF1↑ | ACC↑ | MF1↑ | ACC↑ | MF1↑ |
| Geo | $36.31 \pm 10.15$ | $33.21 \pm 12.25$ | $33.16 \pm 8.32$ | $31.15 \pm 9.25$ | $24.85 \pm 9.43$ | $21.64 \pm 8.94$ |
| Amp | $81.76 \pm 0.89$ | $80.78 \pm 0.78$ | $\mathbf{87.85} \pm 0.83$ | $\mathbf{85.53} \pm 1.10$ | $41.29 \pm 0.56$ | $39.77 \pm 1.03$ |
| Spec | $40.14 \pm 2.05$ | $38.34 \pm 1.95$ | $56.73 \pm 2.01$ | $53.54 \pm 1.98$ | $23.45 \pm 2.55$ | $21.30 \pm 2.41$ |
| Cut | $50.21 \pm 1.34$ | $48.23 \pm 1.23$ | $57.71 \pm 1.12$ | $53.87 \pm 1.09$ | $25.63 \pm 2.95$ | $23.41 \pm 3.11$ |
| Binary | $74.13 \pm 1.12$ | $71.31 \pm 1.10$ | $77.12 \pm 0.75$ | $75.23 \pm 0.95$ | $42.21 \pm 0.97$ | $41.53 \pm 1.10$ |
| Linear | $82.23 \pm 2.10$ | $80.25 \pm 1.93$ | $80.11 \pm 2.05$ | $81.31 \pm 1.73$ | $40.15 \pm 1.43$ | $39.71 \pm 1.14$ |
| Ours | $\mathbf{84.30} \pm 0.73$ | $\mathbf{83.23} \pm 0.58$ | $84.51 \pm 1.10$ | $83.98 \pm 1.03$ | $\mathbf{45.36} \pm 0.97$ | $\mathbf{43.14} \pm 0.81$ |

Table 27: Performance comparison of ours with prior mixups in *Heart Rate Prediction* datasets

| Method | IEEE SPC12 | | IEEE SPC 22 | | DaLia | |
|---|---|---|---|---|---|---|
| | MAE↓ | RMSE↓ | MAE↓ | RMSE↓ | MAE↓ | RMSE↓ |
| Geo | $32.65 \pm 7.25$ | $48.90 \pm 9.87$ | $37.15 \pm 6.74$ | $36.32 \pm 6.21$ | $38.45 \pm 7.31$ | $41.32 \pm 6.21$ |
| Amp | $23.01 \pm 0.95$ | $30.10 \pm 1.04$ | $18.07 \pm 1.13$ | $23.13 \pm 1.43$ | $19.05 \pm 1.63$ | $30.41 \pm 1.65$ |
| Spec | $24.09 \pm 4.10$ | $38.41 \pm 3.98$ | $24.41 \pm 4.10$ | $29.93 \pm 4.10$ | $26.71 \pm 4.34$ | $35.31 \pm 3.93$ |
| Cut | $24.98 \pm 3.93$ | $35.67 \pm 4.15$ | $21.77 \pm 4.45$ | $28.43 \pm 3.97$ | $31.75 \pm 4.10$ | $43.56 \pm 3.88$ |
| Binary | $32.23 \pm 1.67$ | $40.21 \pm 1.98$ | $22.55 \pm 1.87$ | $28.78 \pm 2.10$ | $19.71 \pm 2.15$ | $28.83 \pm 2.45$ |
| Linear | $24.31 \pm 1.54$ | $31.29 \pm 1.75$ | $18.52 \pm 1.43$ | $22.54 \pm 1.49$ | $24.16 \pm 1.89$ | $32.46 \pm 1.97$ |
| Ours | $\mathbf{21.13} \pm 0.89$ | $\mathbf{28.21} \pm 1.15$ | $\mathbf{16.17} \pm 0.85$ | $\mathbf{21.13} \pm 1.05$ | $\mathbf{16.64} \pm 1.20$ | $\mathbf{28.43} \pm 1.43$ |

Table 28: Performance comparison of ours with prior mixups in *CVD classification* datasets

| Method | CPSC 2018 | Chapman |
|---|---|---|
| | AUC↑ | AUC↑ |
| Geo | $45.65 \pm 6.43$ | $61.32 \pm 5.79$ |
| Amp | $\mathbf{84.10} \pm 1.05$ | $89.83 \pm 1.12$ |
| Spec | $69.26 \pm 3.10$ | $70.48 \pm 3.05$ |
| Cut | $72.20 \pm 2.98$ | $79.23 \pm 2.75$ |
| Binary | $80.53 \pm 1.62$ | $82.56 \pm 1.45$ |
| Linear | $78.02 \pm 1.43$ | $90.21 \pm 1.15$ |
| Ours | $83.79 \pm 1.10$ | $\mathbf{93.85} \pm 1.05$ |

## F   Illustrative Examples

In this section, we show examples of the destructive behavior of linear mixup and how our proposed mixup technique solves this problem. In Figure 5 **a)**, we show two PPG waveforms that are obtained from IEEE SPC15 with the same label i.e., the same heart rate value. Also, we give the corresponding frequency domain transformations of these two waveforms in Figure 5 **b)** where the frequency axis is converted to heart rate in beats-per-minute i.e., 1 Hz corresponds to 60 bpm.

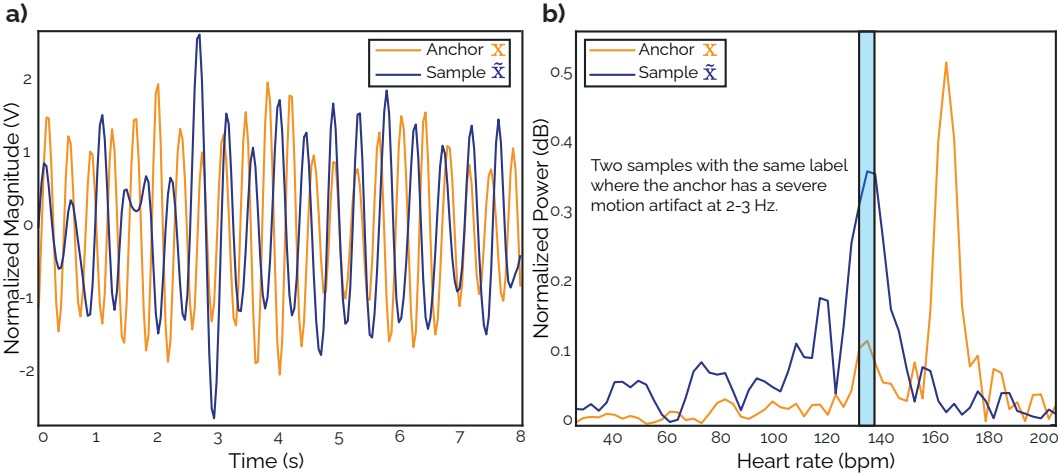

Figure 5: **a)** The waveforms of anchor and random sample, **b)** The frequency domain $(A(\mathbf{x}))$ representation of two samples.

When the linear mixup is applied as in Equation 47 with a $\lambda$ of 0.9, the resulting waveform is anticipated to contain heart rate information to an extent similar to both the anchor and the sample.

$$\mathbf{x}^+ = \lambda\mathbf{x} + (1 - \lambda)\tilde{\mathbf{x}} \tag{47}$$

However, when there is a phase difference greater than $\pi/2$ between these two samples in the frequencies where the task-specific information is carried, the linear mixup destroys the information.

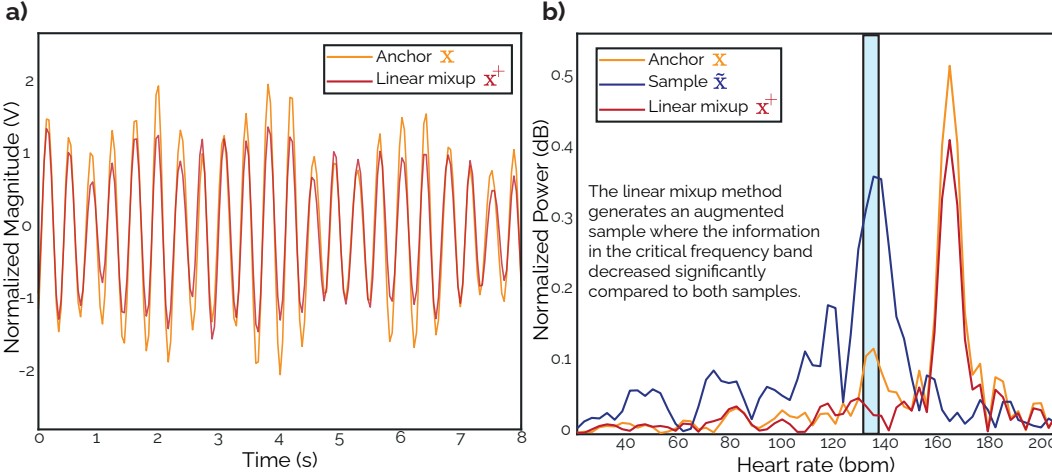

Figure 6: **a)** The waveform of anchor and augmented sample with linear mixup, **b)** The frequency domain $(A(\mathbf{x}))$ representations of samples where the augmented waveform has lost all the information in the critical frequency band, i.e., the task-specific information is lost.

Figures 5 and 6 demonstrate the destructive behavior of linear mixup instead of feature interpolation. The linear mixup technique destroys the task-specific information even though the two samples have the same labels and the mixup ratio is relatively high.

As our proposed mixup prevents this problem and interpolates between features of two samples, the information is not lost but rather enhanced as both samples have the same label, shown in Figure 7.

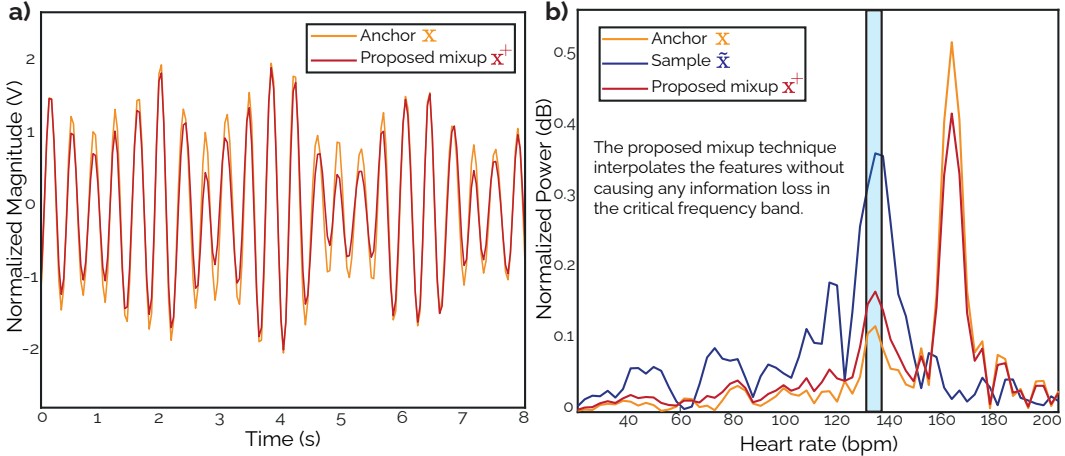

Figure 7: **a)** The waveform of anchor and augmented sample with proposed mixup technique, **b)** The frequency domain $(A(\mathbf{x}))$ representations of samples where the augmented waveform carries the information in the critical frequency band as an interpolation of two samples.

As can be seen From figures 6 and 7, our proposed mixup technique not only prevents information loss due to linear mixup but also generates an interpolated sample.

# G Performance in Supervised Learning Paradigm

We also conduct experiments in the supervised learning paradigm with our proposed mixup method to see its effectiveness in different learning paradigms. We compare the performance of our method with prior mixup techniques. During the experiments, we follow the original implementation where the mixup is applied to the same minibatch after random shuffling. In the seminal work of mixup [31], the authors stated that interpolating only between inputs with equal labels does not lead to performance gains. Therefore, we only perform the tailored mixup without implementing any VAEs. We implement the tailored mixup for the supervised learning as in Equation 48. Although the phase equations can be simplified by omitting the absolute value, we leave it similar to Equation 4 for consistency.

$$\mathbf{x}^+ = \mathcal{F}^{-1}(A(\mathbf{x}^+)\angle P(\mathbf{x}^+)) \quad \text{where} \quad A(\mathbf{x}^+) = \lambda_A A(\mathbf{x}) + (1 - \lambda_A)A(\tilde{\mathbf{x}}) \quad \text{and}$$

$$P(\mathbf{x}^+) = \begin{cases} P(\mathbf{x}) - |\Delta\Theta| * (1 - \lambda_P), & \text{if } \Delta\Theta > 0 \text{ and } \lambda_A \geq 0.5 \\ P(\mathbf{x}) + |\Delta\Theta| * (1 - \lambda_P), & \text{if } \Delta\Theta \leq 0 \text{ and } \lambda_A \geq 0.5 \\ P(\tilde{\mathbf{x}}) - |\Delta\Theta| * (1 - \lambda_P), & \text{if } \Delta\Theta > 0 \text{ and } \lambda_A < 0.5 \\ P(\tilde{\mathbf{x}}) + |\Delta\Theta| * (1 - \lambda_P), & \text{if } \Delta\Theta \leq 0 \text{ and } \lambda_A < 0.5 \end{cases} \tag{48}$$

$$y^+ = \lambda_A y_{\mathbf{x}} + (1 - \lambda_A)y_{\tilde{\mathbf{x}}},$$

where the coefficient for the $\lambda_A$ is chosen from a beta distribution with $\alpha \in [0.1, 0.4]$ within the same range of the original implementation [31]. The mixing for the phase is constrained to our original implementation with a uniform $\lambda_P \sim U(0.9, 1)$. We searched for the best $\alpha$ value for each time-series task and augmentation method. Unlike linear mixup and our proposed approach, for cutmix, we followed the recommendation from the original paper and searched the $\alpha$ value close to 1.

Table 29: Performance comparison in *Activity Recognition* within supervised learning scheme

| Method | UCIHAR | | HHAR | | USC | |
|---|---|---|---|---|---|---|
| | ACC↑ | MF1↑ | ACC↑ | MF1↑ | ACC↑ | MF1↑ |
| W/o Augs. | $65.66 \pm 0.23$ | $61.21 \pm 0.15$ | $91.58 \pm 0.07$ | $91.64 \pm 0.11$ | $71.93 \pm 0.54$ | $68.43 \pm 0.78$ |
| Linear Mix | $77.06 \pm 0.18$ | $73.21 \pm 0.17$ | $93.64 \pm 0.17$ | $93.67 \pm 0.08$ | $74.45 \pm 0.28$ | $71.93 \pm 0.43$ |
| Amp Mix | $70.96 \pm 0.19$ | $67.14 \pm 0.33$ | $92.50 \pm 0.15$ | $92.54 \pm 0.10$ | $74.02 \pm 0.19$ | $71.90 \pm 0.26$ |
| Binary Mix | $69.01 \pm 0.36$ | $71.63 \pm 0.11$ | $92.36 \pm 0.19$ | $92.42 \pm 0.10$ | $72.81 \pm 0.15$ | $70.98 \pm 0.35$ |
| CutMix | $67.14 \pm 0.54$ | $63.31 \pm 0.48$ | $90.37 \pm 0.43$ | $90.36 \pm 0.76$ | $57.89 \pm 0.34$ | $61.45 \pm 0.57$ |
| Ours | $\mathbf{81.60} \pm 0.15$ | $\mathbf{79.35} \pm 0.13$ | $\mathbf{94.02} \pm 0.05$ | $\mathbf{94.00} \pm 0.06$ | $\mathbf{74.85} \pm 0.19$ | $\mathbf{72.45} \pm 0.34$ |

Table 30: Performance comparison in *Heart Rate Prediction* within supervised learning scheme

| Method | IEEE SPC12 | | IEEE SPC 22 | | DaLia | |
|---|---|---|---|---|---|---|
| | MAE↓ | RMSE↓ | MAE↓ | RMSE↓ | MAE↓ | RMSE↓ |
| W/o Augs. | $20.01 \pm 0.03$ | $27.16 \pm 0.05$ | $20.29 \pm 0.87$ | $26.60 \pm 1.13$ | $6.58 \pm 0.10$ | $11.30 \pm 0.58$ |
| Linear Mix | $20.07 \pm 0.09$ | $26.93 \pm 0.10$ | $19.98 \pm 0.12$ | $24.90 \pm 0.51$ | $6.97 \pm 0.14$ | $12.07 \pm 0.51$ |
| Amp Mix | $20.14 \pm 0.07$ | $26.98 \pm 0.07$ | $19.61 \pm 0.07$ | $\mathbf{24.11} \pm 0.21$ | $11.20 \pm 0.17$ | $16.07 \pm 0.43$ |
| Binary Mix | $21.05 \pm 0.13$ | $27.02 \pm 0.08$ | $19.62 \pm 0.10$ | $25.23 \pm 0.13$ | $7.35 \pm 0.16$ | $12.17 \pm 0.53$ |
| CutMix | $20.12 \pm 0.06$ | $\mathbf{26.89} \pm 0.11$ | $19.64 \pm 0.13$ | $24.18 \pm 0.20$ | $10.78 \pm 1.23$ | $14.40 \pm 1.43$ |
| Ours | $\mathbf{19.97} \pm 0.05$ | $26.98 \pm 0.10$ | $\mathbf{19.45} \pm 0.12$ | $24.35 \pm 0.18$ | $\mathbf{6.49} \pm 0.08$ | $\mathbf{11.69} \pm 0.10$ |

Table 31: Performance comparison in *CVD classification* within supervised learning scheme

| Method | CPSC 2018 | Chapman |
|---|---|---|
| | AUC↑ | AUC↑ |
| W/o Augs. | $82.01 \pm 0.51$ | $92.27 \pm 0.35$ |
| Linear Mix | $80.29 \pm 0.93$ | $93.02 \pm 0.33$ |
| Amp Mix | $80.01 \pm 0.36$ | $89.11 \pm 0.27$ |
| Binary Mix | $78.10 \pm 0.98$ | $80.31 \pm 0.36$ |
| CutMix | $80.75 \pm 0.78$ | $89.17 \pm 0.58$ |
| Ours | $\mathbf{83.75} \pm 0.32$ | $\mathbf{95.26} \pm 0.24$ |

