# OpenReview forum: "Finding Order in Chaos: A Novel Data Augmentation Method for Time Series in Contrastive Learning"
_NeurIPS.cc/2023/Conference — NeurIPS 2023 poster_

### Official Review · Reviewer_CSD8 · 2023-07-03

**Soundness:** 3 good
**Presentation:** 2 fair
**Contribution:** 3 good
**Rating:** 5
**Confidence:** 5

**Summary:**

This paper presents a new time series augmentation method which adopts Mixup on the amplitude and phase, respectively, after transforming the time series in the frequency domain, thus avoiding the destructive extrapolation from linear Mixup.

**Strengths:**

1. The authors propose a novel mixup method in the frequency domain for time series, avoiding the destructive extrapolation resulting from linear Mixup.

2. The theoretical proof is complete and has some reference value.


**Weaknesses:**

1. It is unclear whether it works when considering amplitude or phase alone for Mixup. The relevant ablation experiments must be performed consistently with the parameter settings of the original method.

2. Lack of convincing examples. For example, to illustrate the problem, perform Mixup on two samples where the frequency domain information in both samples that are strongly correlated with I(x,y) exactly cancels each other and shows whether the newly generated samples have specific frequency domain information that can be identified.

3. The paper's content does not explain how to "Find Order in Chaos" mentioned in the title and how the "chaos" is reflected. And I'm confused about how the so-called "control the degree of chaos" proposed by the authors is done.

4. Assumption 2.1 is too idealistic. The label information of time series is not only related to the time domain but also the frequency domain, and focusing on the frequency domain information only does not achieve the classification well.


**Questions:**

1. The ablation experiments considering amplitude or phase alone for mixup need to be performed consistently with the parameter settings of the original method.
2. Please give practical examples of how Mixup may corrupt the frequency domain information used to discriminate the generated samples.
3. It is not clear how the so-called "control the degree of chaos" proposed by the authors is done.
4. For data augmentation of time series, why does this paper choose to perform augmentation in the frequency domain instead of the time domain? In general, it is difficult to use only frequency domain information for classification over the time domain for time series classification tasks [1]. In contrast, the combination of time and frequency domain information can effectively improve the model's classification performance [2].
5. What are the advantages of performing data augmentation in the frequency domain instead of doing data augmentation in the time domain of time series data?
6. Why are eight datasets chosen for experiments in this paper? Its inconsistency with existing benchmark datasets for time series prediction [3,4,5], classification [6,7] and anomaly detection [8] tasks. In addition, the comparison methods in this paper do not include benchmark methods in the time series domain. For example, benchmark methods for time series prediction include Informer [3], CoST [4], and FEDFormer [5], etc., time series classification methods include OS-CNN [6] and DSN [7], etc., and time series anomaly detection includes Anomaly Transformer [8], etc.

[1] Cross reconstruction transformer for self-supervised time series representation learning. arXiv, 2022.
[2] Self-Supervised Contrastive Pre-Training for Time Series via Time-Frequency Consistency. NeurIPS, 2022.
[3] Informer: Beyond Efficient Transformer for Long Sequence Time-Series Forecasting. AAAI, 2021.
[4] CoST: Contrastive Learning of Disentangled Seasonal-Trend Representations for Time Series Forecasting. ICLR, 2022.
[5] FEDformer: Frequency Enhanced Decomposed Transformer for Long-term Series Forecasting. ICML, 2022.
[6] Omni-Scale CNNs: a simple and effective kernel size configuration for time series classification. ICLR, 2022.
[7] Dynamic Sparse Network for Time Series Classification: Learning What to "See". NeurIPS, 2022.
[8] Anomaly Transformer: Time Series Anomaly Detection with Association Discrepancy. ICLR, 2022.


**Limitations:**

The authors did not mention the limitations. I do not have any comments on this point.

---

> ### Author Rebuttal · Authors · 2023-08-08
>
> We thank the reviewer for the thorough review. We appreciate the feedback and the recognition of the importance of our work. We replied to each concern as follows.
>
> 1)Regarding the ablation experiments focusing on amplitude or phase alone for the mixup. We'd like to clarify that we conducted a parameter search to determine the optimal settings for all mixup methods. This approach is commonly used as a general practice to ensure the best performance of the baselines in different domains.
> However, it's important to note that the original mixup method was initially designed for images with a distinct distribution compared to time series. As such, directly employing it for comparison would be inequitable. Even for that, the parameters for the mixup methods are very close to the original ones in the literature within a range of 0.8 to 0.9.
>
> 2)Linear mixup has the potential to directly distort information across both time and frequency domains, rather than being limited solely to the frequency, as demonstrated by the mathematical proof we provided.
> As a practical example: Assume you've two signals with the same frequency (PPG or ECG with the same heart rate), if the phase difference between these two signals is more than $\pi/2$, there will be a destructive interference and the magnitude of the waves will decrease depending on the phase difference and magnitude of them.
>
> We believe that it is important to mention the mixup is nothing more than adding two signals together with different ratios. If you apply this to quasi-periodic signals, your waves can destroy each other. This is quite common in nature as well from ocean waves to lights where the most famous example is the double-slit experiment [1-2] which is considered evidence for the probabilistic nature of quantum mechanics.
>
> [1]Young,Thomas (1804). "The Bakerian lecture. Experiments and calculation relative to physical optics". Philosophical Transactions of the Royal Society of London.
> [2]Kipnis,Naum S. (1991). History of the Principle of Interference of Light. Springer
>
> The empirical evidence of this destructive mixup can also be seen from the question of reviewer JK7u “Why linear mix-up method performs well in activity recognition task?”
>
> The other two tasks involve strong periodicity as they originate from the human cardiovascular system, causing linear mixup to perform much worse compared to the less periodic data from activity recognition.
>
> 3)The recent paper “Chaos is a ladder” [3] ([18] in the manuscript) showed that the data augmentations create chaos where contrastive learning climbs that ladder. Also, they showed that a good data augmentation should create samples that are similar to those in intra-class.“For example, two different cars become very similar when they are both cropped to the wheels”.
> In this work, we proposed to control the augmentation degree by controlling the mixup coefficients while looking at the semantic similarity of the samples in the latent domain in an unsupervised way. For example, if the two samples are close to each other in the latent domain, the coefficients are more aggressive to make them closer. Although, we know this approach has some limitations, we mentioned its performance improvement and limitation in Section 5.1 Ablation studies.
>
> [3]Yifei Wang et al. Chaos is a ladder:  A new theoretical understanding of contrastive learning via augmentation overlap. ICLR 2022
>
> 4-5)We think these two questions align at the same point and there is a misconception we would like to clarify.
> In this paper, we have introduced a novel approach to overcome the limitations of mixup by shifting the mixing process from the time domain to the frequency. We did this because we want to get rid of the destructive mixup, which is due to the phase difference between the same frequencies. We had to split these two pieces of information for a wave that motivates us to shift the frequency domain. Otherwise, our augmentation method has a direct correspondence within the time domain too.
>
> We believe that the time or frequency domain has no specific advantage over it in data augmentation for contrastive learning. It is just a matter of how you can keep the task-related information while increasing the diversity of samples, which was shown by InfoMin [53,in script], a seminal work in augmentations for contrastive learning.
>
> 6)The selection of 8 datasets for experiments was driven by the objective of our study, which is to investigate the mixup with quasi-periodic signals and their related applications. While benchmark methods like Informer, CoST, and FEDFormer are valuable in their respective domains, they are not employed as benchmarks within the datasets we used. If you look at the previous works with the same datasets, you can see we used the common benchmarks [4-7].
>
> [4]Hangwei Qian et al. Latent independent excitation for generalizable sensor-based cross-person activity recognition. AAAI 2021
>
> [5]Garrett Wilson et al. Multi-source deep domain adaptation with weak supervision for time-series sensor data. ACM SIGKDD 2020
>
> [6]Dwaipayan Biswas et al. Cornet: Deep learning framework for ppg-based heart rate estimation and biometric identification in the ambulant environment. IEEE Transactions on Biomedical Circuits and Systems, 2019
>
> [7]Francisco Javier Ordóñez et al. Deep convolutional and lstm recurrent neural networks for multimodal wearable activity recognition. Sensors, 2016.
>
> If we use the models you suggested for these tasks, a similar question "Why did you use different models for these datasets?" can be asked again.
>
> Regarding assumption 2.1, the intention behind is not to claim that label information is exclusively encoded within the time or frequency domain. Rather, it asserts that the signal-to-noise ratio contains informative characteristics of a wave.
>
> Again, we thank the reviewer for this careful review and appreciation of our work. Hopefully, these answers have clarified any questions you may have had.

---

> > ### Comment · Reviewer_CSD8 · 2023-08-18
> >
> > Thank you for your response. Most of my questions have been resolved. However, I still have some concerns. The examples provided by the authors are mainly focused on evident frequency features, such as electrocardiogram recordings. Nevertheless, time series in the real world also encompasses situations where frequency features are not as apparent，e.g., AllGestureWiimotex, MelbournePedestrian, and GestureMidAirD3 in UCR dataset.

---

> > > ### Author Response · Authors · 2023-08-19
> > > **Thanks for your response!**
> > >
> > > We thank the reviewer for the response but we want to highlight some points.
> > >
> > > In this paper, we showed a problem of mixup in quasi-periodic time-series data through both theoretical and empirical means. Then, we proposed a method that solves this problem and presented our results with 8 datasets and 14 baselines (previous data augmentation methods and several mixup techniques) where our method outperforms baselines in 7 datasets and ranks second in the other dataset.
> > >
> > > We provided examples where the data is mainly quasi-periodic (activity, cardiovascular, etc.) as we solve a problem directly related to that.
> > > There are hundreds of time-series data with different characteristics, we hoped that the reviewers would anticipate that and appreciate the contribution to the field of quasi-periodic signals. Also, we believe that the time series we demonstrate the performance of our method is of huge importance to the field of ML and health such as electrocardiograms, which make the contribution of this work valuable.
> > >
> > > Thank you once again for joining the discussion section.

---

### Official Review · Reviewer_JMVG · 2023-07-04

**Soundness:** 4 excellent
**Presentation:** 3 good
**Contribution:** 4 excellent
**Rating:** 8
**Confidence:** 3

**Summary:**

This paper proposes a novel mixup method for non-stationary time-series data to generate positive samples for the contrastive learning formulation. The proposed method mixes the magnitude and phase of each frequency component of two samples that are close in the latent space of a variational autoencoder. The authors prove that the mixup process in the frequency domain does not cause the loss of information while it generates diverse samples. The paper provides an extensive empirical evaluation to show that the proposed method learns better representations than existing contrastive learning baselines on eight time series datasets from three tasks.

**Strengths:**

The paper studies an important and under-explored problem. Although contrastive learning is proven to work well on images, its performance on time series is yet limited due to the lack of good time series augmentations. While many existing work tries to find the good time series augmentations through extensive empirical studies, this paper proposes a theoretically well-grounded mix-up method for time series augmentation. Moreover, the empirical evaluation is also extensive and convincing.

**Weaknesses:**

1. The hyperlinks of the citations and the references are missed.

2. I don't find where $x^*$ is defined.

3. It would be great if there is an algorithm box explaining the whole process of drawing samples, computing the degree of the augmentation, performing the mixup, and then training.

**Questions:**

1. How do you select $\beta$? Would a very large (close to 1) $\beta$ limit the diversity of the augmented samples?

2. What does the instance selection (line 268) mean exactly?

**Limitations:**

Yes

---

> ### Author Rebuttal · Authors · 2023-08-08
>
> We thank the reviewer for the thorough review of our work. We appreciate the feedback and are glad to hear that our work has been received positively.
>
>
> Regarding the weaknesses,
>
> thank you for bringing hyperlinks to our attention. We will fix this issue in the revised version of the manuscript. Your feedback is greatly appreciated.
>
>
> We appreciate your observation regarding the absence of the definition for "x*." In prior works, namely DACL and GenRep, this term was denoted as the optimum generated or augmented sample. This contextual reference guided our usage of the term before formally defining it. However, we understand the importance of clarity and will provide a precise definition for "x*" in the revised manuscript. We thank you for your valuable suggestion.
>
> We shared the same thought in the submission process but we were keen to add another figure to the manuscript as we have a limited space. However, we appreciate your input and will incorporate a new figure to provide a clear and comprehensive overview of the process in the revisited manuscript or Appendix. Thank you for your thoughtful suggestion.
>
> Regarding the questions, we have replied to each of them below.
>
> Q1) In previous mixup methods normally $\beta$ is chosen with high values such as 0.8 or 0.9. However, the major drawback of this is that when you increase the $\beta$ to closer to 1, the mixup samples are getting closer to the anchor sample, and the diversity of them decreases significantly as you expected. Therefore, we followed a similar procedure with previous works and performed a grid search amongst values 0.7 to 0.9 and chose the best values. We also observed that the $\beta$ values are quite flexible for the magnitude mixing while for the phase it should be more closer to 0.8-0.9. We believe that this can be attributed to the influence of phase values on the semantic characteristics of the signal. Altering these phase values exerts a potent impact on the inherent features conveyed by the signals.
>
> Q2) We thank the reviewer for this careful revision. It is our mistake. It should be the mixup coefficient selection according to phase and magnitude rather than the instance selection as we indeed choose mixup coefficients rather than instances.
>
> By addressing your questions, we hope to provide clarity and resolve any uncertainties you may have. We genuinely appreciate your thorough review and the insights you've shared. Thank you once again for your careful attention to detail during the revision process and appreciation of our work.

---

> > ### Comment · Reviewer_JMVG · 2023-08-19
> >
> > Thank you for addressing my concerns. I maintain my initial recommendation to support the acceptance of the paper. The paper presents a novel time series augmentation approach with applicability across multiple popular contrastive learning frameworks. While the experiments do not encompass some benchmark datasets, the evaluation results across eight datasets unequivocally demonstrate the effectiveness of the proposed method.

---

> > > ### Author Response · Authors · 2023-08-19
> > > **Thanks!**
> > >
> > > We are glad to hear our contribution to the field is appreciated.
> > > Thank you once again for your time and appreciation of our work.

---

### Official Review · Reviewer_hEg3 · 2023-07-06

**Soundness:** 3 good
**Presentation:** 2 fair
**Contribution:** 2 fair
**Rating:** 4
**Confidence:** 4

**Summary:**

This paper is about a novel data augmentation method that can be used in contrastive learning for time-series tasks and aims to connect intra-class samples together, and find order in the latent space. The proposed method builds upon the mixup data augmentation technique by controlling the degree of chaos created by data augmentation. More specifically, the augmentation method considers the phase and the amplitude information as two separate features and then generates positive samples by controlling the mixup coefficients for each feature for each randomly chosen pair. This process helps to generate features that enhance intra-class similarity and help contrastive learning to learn class separated representations. The proposed method is evaluated using three time-series tasks: heart rate estimation, human activity recognition and cardiovascular disease detection and against state of the art comparison methods.

**Strengths:**

- The paper is about an interesting topic, contrastive learning in time-series.
- The proposed methodology is based on the idea of controlling the degree of chaos created by data augmentation methods. These ideas can be applied in the augmentation part of contrastive learning for other domain as well.
- The proposed methodology shows improvement in the performance over ten comparison methods on three tasks and eight datasets. Some time-series contrastive learning methods are missing from the comparison though.

**Weaknesses:**

- The writing in the paper can be improved. The Method and Results and Discussion sections need some organization, adding subsections will help with the structure.
- The proposed method has incremental novelty. This method builds on top of known components. The writing needs improvements to make the contributions of this paper more clear.
- The experimental setup also seems to be missing some important state of the art. No time-series specific contrastive learning method is included in the experiments for comparison.
- The proposed methodology seems to have similarities with TS-TCC [80] and TFC [22] (the reference numbers align with the ones in the main paper) and since these are state of the art in contrastive learning for time-series, they should be part of the comparison methods.
- A comparison of the methodology of the proposed method with [22] and [80] is needed as these paper are around the same ideas. What is different in the contributions of the proposed method?
- Dataset statistics are missing from the Results and Discussion section. Adding them in a table helps with the understanding of the results.

**Questions:**

My main concerns are regarding the readability of the paper and improvements in the structure of the paper, and the experimental setup that is missing time-series contrastive learning models. The results and the discussion will be more convincing and the performance difference more important if the author include the results of some other methods, like [22] and [80]. A comparison of the methodology and the concepts of the proposed method with [22] and [80] is needed as these paper are around the same ideas. What is different in the contributions of the proposed method?

**Limitations:**

There are no limitations or negative societal impact from this work. The datasets used in the paper are all publicly available.

---

> ### Author Rebuttal · Authors · 2023-08-08
>
> We thank you for your comments. And, we would like to clarify some misunderstandings and points arising from your review of our paper.
>
> Regarding the claim of your incremental novelty, it's important to note that there currently exists no prior study that has specifically addressed the destructive effects of the linear mixup when applied to periodic time series data, nor has any previous work proposed a corrective methodology in this context. While linear mixup has demonstrated efficacy within image-related domains, its applicability to time series analysis remains constrained due to the inherent periodic nature of time series data. Our research also highlights that increased signal periodicity shows a significant decline in mixup performance. Given this distinct gap in the existing literature, we argue that our work should not be characterized as incremental in nature.
>
> We believe that our work contributes several insights, including those of technical nature. In summary,
>
> To the best of our knowledge, our work is the first to
>
>
> * present the destructive behavior of linear mixup for quasi-periodic time-series data both empirically and theoretically with mathematical proofs.
> * propose a novel mixup approach for time-series data while preventing information loss of prior works on mixup
> * take a novel approach for sampling mixup coefficients for each pair based on their similarity in the latent space, which is constructed without supervision while learning disentangled representations, to prevent aggressive augmentation between inter-class samples.
> * demonstrate that our approach significantly outperforms well-known mixup methods with state-of-the-art data augmentation techniques in 8 datasets while comparing with 14 baselines.
>
>
> Although we will more strongly emphasize our contributions in the revised version of the paper, we kindly wish to draw the reviewer's focus to the non-incremental nature of the paper's contributions.
>
>
>
> Regarding your concern for [80] and [22], these papers proposed an unsupervised time-series representation learning framework. Therefore, they are similar to SimCLR and BYOL, but not our presented method.
>
> In this paper, we propose a data augmentation strategy to improve the performance of unsupervised contrastive learning frameworks by using a tailored mixup method for quasi-periodic signals. The mixup is well-appreciated in the vision community as it offers an easy and effective data augmentation strategy, even so, many works proposed different variations of mixup such as cutmix, binarmix, geomix. In this paper, we showed that the original mixup and variations do not perform well with the quasi-periodic signals and proposed a tailored theoretically proven mixup method and showed the improvement.
>
>
> To explain further, you can use our proposed method with any unsupervised contrastive framework such as SimCLR, BYOL, NNCLR, MoCo, or [80] and [22] as a data augmentation technique. Therefore in Appendix E, we showed how our proposed data augmentation increases the performance of BYOL and SimCLR compared to the traditional augmentation used in time series such as jittering. However, you cannot use [80] and [22] together with SimCLR or BYOL. They constitute a replacement for those self-supervised learning methods. Therefore, we think our experimental setup with the additional experiments has no missing parts and clearly indicates the contribution of the paper.
>
> Moreover, [22] proposed a pre-training strategy while considering the time-frequency consistency (TF-C) —embedding a time-based neighborhood of an example close to its frequency-based neighborhood, which they mainly used for domain adaptation whereas, in our paper, we showed the destructive feature of mixup for quasi-periodic signals and fix this drawback to increase the performance for unsupervised contrastive learning.
>
> Similarly, in [80]  authors propose a framework to learn time-series representation from unlabeled data. Their framework requires weak and strong data augmentations where they implemented weak augmentation as a jitter-and-scale and strong augmentation as a permutation-and-jitter. We both compared the performance of those augmentations with ours in two different contrastive learning frameworks and showed the improvement.
>
>
> Responding to your question, we have also introduced another contrastive framework explicitly tailored for time series data. In the submitted pdf, we demonstrate the performance enhancement achieved through the integration of our method with this additional framework. Therefore, we can confidently say that our proposed method exhibits substantial distinctions from [80] and [22] in terms of its contributions. Our work is not a time-series contrastive learning method where we should compare with other frameworks as opposed to your claim. Furthermore, it's worth noting that we have conducted comparisons with all previously published known data augmentation methods in 8 datasets.
>
>
> Regarding the dataset statics, we already mentioned some statistics about datasets in Appendix B and we will certainly address this suggestion by incorporating dataset statistics in a table.
>
>
> We apologize for any confusion caused by the misinterpretation. We believe that these clarifications with additional results will now help address any misconceptions and make the contribution of our paper clear to the field.
>
> Thank you for your time and consideration again.

---

> > ### Comment · Reviewer_hEg3 · 2023-08-22
> >
> > Thank you for your response. After reading the other reviews and the rebuttal responses, I understand that the focus of this work is the data augmentation aspect and not a new contrastive learning model (and that was my understanding from the beginning), but I believe that in order to compare the proposed data augmentation strategy it is crucial to try it with the other state of the art contrastive learning models. It is very useful, that the readers who read this paper, would have an idea of what works best with this augmentation strategy and what does not. Also, since the focus of this work is the data augmentation strategy, it would be interesting to check how it works in all domains of supervised, self-supervised and unsupervised learning. Finally, one of the main reasons that we discuss about the focus of this work (data augmentation strategy vs. contrastive learning model) is the way the authors present their findings. Instead of using the names of the contrastive learning models, they could use the same contrastive learning model and only change the augmentation strategies. Then, they can present for example, the comparison of the methods not by using the names of the contrastive learning models, but instead by using the names of the data augmentation strategies. And they could also add one table with the data augmentation strategies in the state of the art and the corresponding models that those have been used in (i.e., table with two columns, one is model's name, one is augmentation strategy description). Overall, I do believe that this paper will make a stronger impact if both a comparison for the data augmentation strategies and the contrastive learning models will be included, as this is one of the mechanisms that researchers use the data augmentations to learn the time-series representations. As a result, I will keep my initial score as I am not fully convinced about the current state of the experimental setup.

---

### Official Review · Reviewer_JK7u · 2023-07-08

**Soundness:** 3 good
**Presentation:** 2 fair
**Contribution:** 2 fair
**Rating:** 4
**Confidence:** 4

**Summary:**

This paper explores data augmentation for contrastive learning that better aligns with the nature of time series. Specifically, they tailor the mix-up method separately to amplitude and phase in the frequency domain. A theoretical analysis illustrates how the method enhances task-specific information, contrasting it to the linear mix-up method which potentially discards task-specific information. Experiments on time-series from three domains also demonstrated the advantages compared with previous mix-up and contrastive learning methods.

**Strengths:**

1. This paper studies an important problem of designing augmentation for contrastive learning that adapts to the non-stationarity of time series.

2. The paper presents theoretical analysis of the benefits of proposed mix-up method compared with vanilla linear mix-up method.

3. The paper shows better performance on time-series datasets of multiple domains.

**Weaknesses:**

**Lack of baseline comparisons**: The paper compares with a list of mix-up methods and contrastive learning methods, and the authors also incorporate time-series augmentation methods into the framework in appendix. However, it would also be interesting to have (1) some direct comparisons with existing time-series augmentation methods, e.g., [1, 2] (both have mix-up augmentation in the time or frequency domain). (2) Contrastive learning methods dedicated for time-series domain [3, 4, 5, 6] (I am curious why [4] is not one of the baselines in experiments?)

[1] FrAug: Frequency Domain Augmentation for Time Series Forecasting

[2] Towards Diverse and Coherent Augmentation for Time-Series Forecasting

[3] CoST: Contrastive Learning of Disentangled Seasonal-Trend Representations for Time Series Forecasting

[4] Self-supervised contrastive pre-training for time series via time-frequency consistency

[5] Unsupervised Representation Learning for Time Series with Temporal Neighborhood Coding

[6] TS2Vec: Towards Universal Representation of Time Series


**Experimental analysis**: Why linear mix-up method performs well in activity recognition task? Is there any difference in the setting of activity recognition compared with the other two tasks?

**Writing**:

1. Consider defining notations of x, \tilde{x}, x^* before or in Proposition 2.3.

2. The title metions "order" and "chaos"; hence, a detailed explanation of these terms, as well as how the proposed method comprehensively accounts for the "non-stationarity" of time series, would be beneficial.

**Questions:**

See weakness above.

**Limitations:**

The paper did not explicitly discuss their limitations. Some comments and questions regarding limitations of the work are discussed in the weakness and question parts above.

---

> ### Author Rebuttal · Authors · 2023-08-08
>
> We appreciate your feedback about our paper on data augmentation for contrastive learning in the context of quasi-period time series data. We also thank you for valuing the importance of the problem. We would like to address certain points raised in your review.
>
> We are well aware of all the works you cited in your review, except the [2].
>
> We apologize for missing reference [2], which emerged close to our submission. This paper indeed proposed a data augmentation method for time series but the overall focus was for forecasting without considering the quasi-periodic nature of signals. We applied their proposed method.But,as they emphasize different frequency components via random weights, its performance is quite low compared to ours (as can be seen from the submitted single-page pdf). We will include them in our revisited version.
>
> Secondly, we haven't explicitly compared our proposed augmentation with [1] in the original script because the work presented in [1] is for time-series forecasting and the augmentations are tailored for this purpose (not for quasi-periodic signals, similar to [2]) where comparison will be unfair. Even in the augmentation, there are look-back window x and target horizon y. How can you think integrating these augmentations to our problems make sense? For example, what is the look-back window and target horizon for ECG signals in cardiovascular disease detection?
>
> If you look at the baselines we used, they all claim their proposed methods can be applied to other domains (e.g.,DACL). However,as a reply to your question, we applied the data augmentations in FrAug (with original parameters) which are mainly frequency masking and mixing, you can find the results in the pdf. The performance of this method is low since sometimes the key freq. components are masked or mixed as the characteristic of quasi-periodic signals is different.
>
> Thirdly, regarding the other cited works, we've already applied the augmentations mentioned there in our experiments. If you look at the augmentations used in these works, you can see we already used and pointed out them in Appendix.
>
>
> [3]CoST:Contrastive Learning of Disentangled Seasonal-Trend Representations for Time Series Forecasting
>
> Scaling,Shifting,Jittering,
>
> All of them are used and given in the submitted manuscript.
>
> [4]Self-supervised contrastive pre-training for time series via time-frequency consistency
>
> In this work, the author's main contribution is not a data augmentation method but a framework for domain adaptation.That was the main reason, we did not explicitly compare this as a baseline, which will not be a fair comparison. In other words, this work is not a baseline for us. We propose a data augmentation method for unsupervised learning, not a framework.
>
> The authors propose to perturb only the magnitude of the frequency domain without changing the phase while the perturbation is performed by sampling from Gaussian. In the paper, the authors showed that perturbation of high and low frequency affects performance differently depending on the application. Therefore, we applied this to the whole band during our submission. As per your request, we also applied to the low band separately as in the original paper. In the end, the performance decreased severely.
>
> [5]Unsupervised Representation Learning for Time Series with Temporal Neighborhood Coding.
>
> There is no data augmentation used or proposed in this paper. Again, we propose a data augmentation method for quasi-periodic nonstationary signals. This work is not even a baseline for our work.
>
> [6]TS2Vec:Towards Universal Representation of Time Series
>
> Timestamp Masking, Random Cropping
>
> We both applied these augmentation techniques as baselines (random zero out and permute)
>
>
> Q)Experimental analysis:Why linear mix-up method performs well in activity recognition task?Is there any difference in the setting of activity recognition compared with the other two tasks?
>
>
> The answer to this question outlines the main contribution of this paper. The mixup method demonstrates a destructive impact in cases where it is applied to two sinusoids with the same frequencies yet different phases. The PPG (for heart rate) and ECG (for cardiovascular disease) signals exhibit a higher degree of quasi-periodicity, leading to diminished performance when used with linear mixup. These two signals display a greater level of periodicity as they stem from the human body. In contrast, in the realm of activity recognition, the extent of periodicity varies with different activities—like sitting (characterized by lower periodicity) and walking (exhibiting higher periodicity).
>
> Our proposed method addresses this issue and leads to a substantial enhancement in performance. Hope, the answer to this question clarifies the contribution of our presented work.
>
> Regarding your claim of lack of baseline comparisons, in our work, we used 8 datasets in 3 domains with 14 baselines including automatic data augmentation techniques (DACL, GenRep) and 6 different mixup methods in 2 contrastive learning frameworks. Our theoretically well-grounded mix-up overperformed all of these in 7 datasets from 3 domains while ranking second in one dataset. Your claim of lack of baseline is not realistic at all. Although we agree that using different contrastive frameworks can be more interesting, our experiments including ablation studies are comprehensive and robust invalidating any notion of inadequate baseline comparisons. Additionally, we include one more contrastive learning framework that is designed for time series where we observe the same behavior.
>
> Hope, the reviewer would be convinced by the additional results and understand the difference between a lack of baseline and more interesting experiments. We believe our work has enough baseline, evaluating with 2 more contrastive learning frameworks does not add anything significant to the already proved contribution that is substantiated both theoretically and empirically.

---

> > ### Comment · Reviewer_JK7u · 2023-08-19
> >
> > Thank the authors for the responses and additional experiments. After reading the rebuttal, I still have a few questions.
> > 1. The authors said "But,as they emphasize different frequency components via random weights, its performance is quite low compared to ours (as can be seen from the submitted single-page pdf)", but I don't see the results of reference [2] in the pdf.
> > 2. I understand that this paper emphasizes improved mix-up methods in contrastive learning for quasi-periodic time-series data, but that does not stand as a valid point for excluding comparison with other types of contrastive learning frameworks. Users are in search for better contrastive learning methods, regardless of the underlying frameworks. If the comparison has to be limited to other augmentation methods, it would be better to generalize the scope to broader settings beyond contrastive learning (e.g., supervised classification and forecasting), and show the merits of the proposed augmentation in general.

---

### Author Rebuttal · Authors · 2023-08-09

We would like to thank all the reviewers for taking the time to read our paper and provide feedback.

In response to the reviewers' comments, we included two more data augmentations for comparison. We also applied our proposed data augmentation method to one more contrastive learning framework, which is specifically proposed for time-series data, in addition to SimCLR and BYOL.

Following the submission of our responses, we would greatly appreciate receiving additional feedback from the reviewers, particularly from Reviewer JK7u and hEg3 (the first two reviewers). Their reviews appear to misinterpret the paper, viewing it more as a novel unsupervised contrastive learning framework rather than recognizing it as a novel data augmentation method for quasi-periodic signals, which is evident from their request for a direct comparison (baseline) with unsupervised learning methods.

Even if the reviewers maintain their initial assessments, we value the opportunity to gain additional clarity on your feedback.

If you believe there are areas where our responses did not fully address your concerns or if there are aspects that could be improved to merit a higher score, we would greatly appreciate your insights on those specific points. Your feedback is crucial in helping us enhance our work moving forward.

Thank you once again for your review!

---

### Decision · Program_Chairs · 2023-09-21

**Decision:**

Accept (poster)

**Comment:**

This paper introduces a well-motivated data augmentation strategy for continuous time series data. A data modality often understudied by mainstream ML conferences, while being one of the most prevalent types of data sources in the real world. This work is relevant for whole fields focusing on modeling or predicting based on biosignals.

The paper is a great example of how new data augmentation strategies should be introduced, instead of just trying out a random strategy, the authors thoroughly motivate and validate their proposed method and then show how it leads to meaningful improvements on a wide range of tasks.

There are two remaining concerns of borderline reject reviewers which I will address below:

1. STAug baseline missing

This indeed seems to be missing from the author response pdf, please make sure to include these numbers in the final version of the manuscript. This oversight does not diminish the careful motivation and analysis of the method.

2. Request to compare to more contrastive learning methods.

It is sufficient to focus on the most important well-established methods combined with the propose augmentation strategy. The empirical validation in this paper is extensive and I consider comparing to even more baselines out of scope for this work. The authors argued extensively to substantiate why comparing to more self-supervised learning methods will not benefit the current paper and I agree with them.